# Bridging Non-Intrusive Tracing and Fine-Grained Cross-Layer Representations for LLM Inference Diagnosis

## Abstract

LLM inference spans the inference engine, compute backend, host operators, and GPU kernels, where asynchrony and concurrency make request-level end-to-end observability and diagnosis challenging. We present TRUFFLD, a non-intrusive and cross-layer framework that provides fine-grained representations for diagnosis in large-scale LLM inference. For data collection, TRUFFLD activates NVTX markers and CUPTI callbacks to capture raw events from vertical (intra-node stack execution) and horizontal (cross-node communication) perspectives. We then propose a call-chain merging algorithm that aligns these events on a unified time base and reconstructs a per-request call-chain tree preserving both structural and temporal semantics. For anomaly detection, TRUFFLD adopts a two-stage pipeline. A Gaussian Mixture Model models multi-modal normal behavior and produces calibrated numeric confidences, while a large language model applies structure- and context-aware reasoning to generate step-level decisions and operator-level localization. Experiments on a multi-node GPU cluster running `Qwen-8B` inference with both online and offline workloads demonstrate near-perfect step-level detection and superior operator-level performance compared to multiple baselines, with low deployment overhead and no modification to binaries. TRUFFLD provides a practical end-to-end solution for observability and diagnosis in large-scale LLM inference.

## 1 Introduction

Large language models (LLMs) have achieved remarkable progress across natural language processing and multimodal tasks, and online inference services are consequently being deployed at scale and operated continuously.(Radford et al., 2018; Raffel et al., 2020; Brown et al., 2020; Touvron et al., 2023; Achiam et al., 2023; Bai et al., 2023; Gemini Team et al., 2023; Gemma Team et al., 2024) A typical LLM inference stack spans multiple layers: the *inference engine layer* (scheduling and batching, e.g., continuous batching), the *compute backend layer* (tensor operators and memory management, e.g., PyTorch or specialized libraries), the *host-operator layer* (CPU-side CUDA runtime/driver calls), and the *device-operator layer* (GPU kernels and memory operations). (Kwon et al., 2023; Jiang et al., 2023) When such a multilayer stack runs under high concurrency on heterogeneous hardware, small perturbations at any layer can be amplified along the chain, leading to tail-latency inflation, throughput degradation, and instability.(Bechtel & Yun, 2023; Di et al., 2016; Olston et al., 2017) This makes **request-level, end-to-end observability and anomaly localization** a core requirement for operations and performance optimization.

To realize request-granular, cross-layer, and interpretable tracing and diagnosis in production LLM inference, several challenges must be addressed. First, event collection and correlation need to be performed in a non-intrusive manner, without modifying online services or model binaries, so as to avoid secondary impact on latency and stability. Second, host-side operator launches and device-side kernel executions are inherently asynchronous, and their timestamps differ across domains, which requires precise cross-domain alignment and range matching to represent the same logical operator consistently. Third, multiple requests often execute concurrently, and coroutine or batch schedulers may merge them into a single input matrix, producing many-to-one operator invocations; reliable request-level attribution therefore demands that sufficient semantic context be retained at collection

time. Finally, heterogeneous events from the engine, backend, host, and device layers must be reconciled into a consistent request-level call chain, enabling complete end-to-end attribution and localization.

To address these challenges, we propose TRUFFLD—A Non-Intrusive Tracing and Fine-Grained Representations Unified Framework For LLM Diagnosis. TRUFFLD records raw events from *vertical* (within-node stack execution) and *horizontal* (cross-node communication) views using NVTX and CUPTI without modifying online binaries. It then normalizes timestamps, reconstructs nested ranges, and aligns host and device by identifiers to build a request-level call-chain tree that covers engine, backend, host, and device. On top of this representation, TRUFFLD performs anomaly detection with a two-stage pipeline: a Gaussian Mixture Model that learns multi-modal normal behavior and outputs calibrated confidences, followed by a large language model that applies structure- and context-aware reasoning to decide abnormal steps and localize faulty operators.

Our contributions are as follows.

**(i) Non-intrusive monitoring with low overhead.** TRUFFLD records execution events without modifying source code or online binaries, ensuring that tracing introduces minimal interference in production.

**(ii) Fine-grained cross-layer representations.** By jointly capturing vertical (per-node stack execution) and horizontal (cross-node communication) views, TRUFFLD provides comprehensive coverage and enables per-step granularity at the request level.

**(iii) Asynchrony- and concurrency-aware call-chain construction.** TRUFFLD reconstructs request-level call-chain trees by aligning host and device timelines, and by explicitly disambiguating many-to-one operator mappings caused by batching and coroutine scheduling.

**(iv) End-to-end anomaly diagnosis.** On top of the call-chain representation, TRUFFLD integrates a probabilistic stage based on Gaussian Mixture Models to model multi-modal normal behavior, and a reasoning stage based on large language models to apply structural and semantic constraints. This two-stage design yields both step-level anomaly decisions and operator-level localization, and outperforms classical machine learning baselines.

By introducing TRUFFLD, we substantially enhance observability for LLM inference and provide a principled framework for anomaly diagnosis in production inference. Our implementation is available in the supplemental material for double-blind review.

## 2 PRELIMINARIES

### 2.1 SYSTEM STACK IN LLM INFERENCE

A typical LLM inference system can be decomposed into five abstract layers along the software–hardware stack.

**(i) Inference engine layer.** Maps incoming requests into tensor batches and manages KV-cache, scheduling, and parallelization strategies (e.g., continuous batching with distinct prefill and decoding phases, as in modern engines such as `vLLM`).(Kwon et al., 2023)

**(ii) Compute backend layer.** Provides tensor operators and memory management. PyTorch is the default in many deployments, while specialized libraries (e.g., xFormers, FlashAttention) can be enabled via environment configuration.(Paszke et al., 2019; 2017; Dao et al., 2022)

**(iii) Host-operator layer (CPU side).** Invokes the CUDA runtime and driver as well as user-space libraries to launch kernels, perform memory copies, and manage streams and events.(NVIDIA, 2024a)

**(iv) Device-operator layer (GPU side).** Executes kernels and memory operations on the accelerator, exposing SM utilization, memory bandwidth pressure, and warp scheduling dynamics.(Stephenson et al., 2015)

Two properties of this stack are critical for tracing. *Asynchrony* arises because host launches (e.g., `cudaLaunchKernel`) and device execution have misaligned timestamps, requiring cross-domain reconciliation. *Concurrency* arises because schedulers merge requests into shared operators and

communication synchronizes across devices, creating many-to-one and cross-node dependencies. These factors complicate request-level attribution and demand precise context propagation and timeline alignment.

## 2.2 RELATED WORK

**AI/ML performance profiling and diagnosis.** System profilers such as Nsight Systems and PyTorch Profiler are widely used to analyze training and inference, exposing operator timelines, CUDA activity, and selected framework events (NVIDIA, 2025b; PyTorch, 2025a; TensorFlow, 2020). eBPF-based observability has progressed as well, including OpenTelemetry eBPF profilers that capture kernel and some user-space events with low overhead (Xu et al., 2025; Zadeh et al., 2023; OpenTelemetry, 2024). In distributed settings, work on NCCL tuning, workload scheduling, and topology planning targets communication efficiency and scalability (Sigelman et al., 2010; OpenTracing, 2024). Despite these advances, request-level, cross-layer, end-to-end profiling for LLM inference remains limited, since batching and host–device asynchrony complicate consistent call-chain reconstruction.

**NVTX and CUPTI.** NVTX provides lightweight ranges and marks across C, C++, and Python, while CUPTI offers callbacks and activity records for CUDA runtime, driver, and kernel execution under a unified time base (NVIDIA, 2024c;b). The two are naturally complementary and support precise host–device alignment without modifying application source, which suits non-intrusive cross-layer tracing in modern LLM inference stacks.

**Failure localization.** Classical methods learn from telemetry using clustering or density modeling, for example KMeans, DBSCAN, Isolation Forest, and GMM for unsupervised anomaly detection (MacQueen, 1967; Ester et al., 1996; Breiman, 2001). Recent systems employ LLMs to read structured summaries and logs and to infer root causes under light rules in cloud and network operations (Ahmed et al., 2023; Chen et al., 2024; Jin et al., 2023; Roy et al., 2024; Wang et al., 2024). Statistical models provide stable numeric signals, whereas LLMs integrate heterogeneous context and weak structural cues.

Our TRUFFLD combines these lines of work by using NVTX and CUPTI to collect non-intrusive cross-layer events, then reconstructing a unified request-level call-chain tree that spans engine, backend, host, and device. On this representation, a two-stage diagnosis pipeline first fits Gaussian Mixture Models to obtain calibrated confidences and then applies an LLM to incorporate structural and semantic context for step-level decisions and operator-level localization. The method closes the gap between low-level traces and actionable end-to-end diagnosis. Details follow in §3.

## 3 METHOD

### 3.1 SYSTEM ARCHITECTURE

TRUFFLD targets production LLM inference with three design goals: request-level end-to-end coverage, non-intrusive deployment, and high-fidelity cross-layer tracing. As shown in Fig. 1, our framework consists of two complementary components that together deliver observability and diagnosis at the step and operator levels.

**Event collection and call-chain construction.** The first component gathers execution signals along the inference path, using NVTX and CUPTI to record raw events from *vertical* (per-request stack execution) and *horizontal* (cross-node communication) perspectives. These events are aligned on a unified time base and merged into per-request call-chain trees that preserve both timing and semantics. This component addresses *how to obtain reliable data with minimal perturbation*.

**Two-stage anomaly detection.** The second component consumes the structured call-chains and performs diagnosis in two stages. A *probabilistic stage* fits Gaussian Mixture Models to operator features, producing calibrated numeric confidences that quantify normality at both instance and family levels. A *reasoning stage* then integrates these confidences with structural context and lightweight semantics (e.g., operator names, backend tags, kernel families, communication size, and phase markers) to yield step-level decisions and operator-level localization. This component addresses *how to turn traces into effective and explainable diagnosis*.

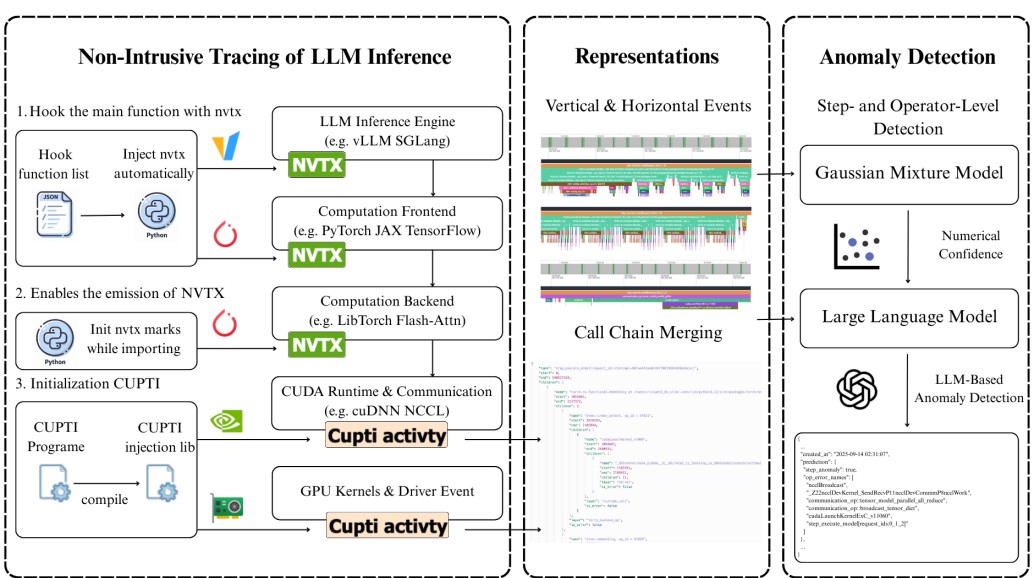

Figure 1: System overview of TRUFFLD. The framework integrates event collection and call-chain construction (left, middle) with a two-stage anomaly detection pipeline (right), enabling request-level observability and diagnosis at both step and operator levels.

## 3.2 DATA COLLECTION

Fine-grained and accurate data collection is the foundation for downstream tasks such as anomaly detection and diagnosis. We therefore design a two-stage process. The first stage exploits the layered structure of the LLM inference stack, recording raw events from vertical (per-request stack execution) and horizontal (cross-node communication and synchronization) perspectives. The second stage takes these raw events as input and reconstructs structured call chains that serve as the basis for subsequent analysis in §3.3.

### 3.2.1 LAYERED SIGNAL ACQUISITION

**Inference engine.** Concurrent batching in `vLLM` interleaves requests across decoding iterations. We dynamically inject an NVTX range at the execution entry and treat each iteration as an observation unit; the NVTX message encodes the set of requests present in the batch. Nested NVTX semantics propagate this request context to lower layers, so iteration-level collections remain attributable at the request level even under interleaving.

**Computation front end and backend.** Front-end operators execute in Python, while computation is realized in C++/CUDA. We enable framework NVTX marks to recover backend operator tags and lightly wrap a small set of front-end functions to record call sites and minimal context. This establishes a stable correspondence between front-end operators and backend execution across the language boundary.[1]

**CUDA API and GPU kernels.** Host and device execute asynchronously. Each worker process loads CUPTI so that runtime calls, driver calls, GPU kernel executions, and Python-level NVTX marks are captured on the same time base. CUPTI correlation identifiers pair each kernel with its launching host call, which is then located inside the enclosing NVTX range, yielding a consistent path from device kernels to engine semantics.

**Communication.** Distributed inference introduces collective operations that often overlap with computation. We mark engine communication operations with NVTX ranges and combine them with CUPTI runtime and kernel events to recover structured communication chains linking request

---

[1]The table of injected NVTX hooks for the PyTorch front end is provided in Appendix A.

context, engine dispatch, NCCL operators, host launches, and device kernels. Stream identifiers and timestamps separate overlapped compute and communication on a common time axis.

This layered strategy yields two raw event sets: a vertical set describing per-request stacks and a horizontal set describing per-step communication, which are fed into the following call-chain merging algorithm in §3.2.2. Representative raw-event excerpts are provided in Appendix B.

### 3.2.2 CALL-CHAIN MERGING

All raw events are first normalized onto a single nanosecond time base, ensuring consistency across host and device domains. Each event is represented as a tuple

$$e = (\mathsf{type}, \mathsf{name}, \mathsf{pid}, \mathsf{tid}, \mathsf{id}, \mathsf{cid}, \mathsf{fields}),$$

where $\mathsf{type} \in \{\mathsf{NVTX\_PUSH}, \mathsf{NVTX\_POP}, \mathsf{RUNTIME}, \mathsf{DRIVER}, \mathsf{KERNEL}\}$, and $\mathsf{id}, \mathsf{cid}$ are optional NVTX and CUPTI identifiers. Define a unified timestamp $\tau : \mathcal{E} \to \mathbb{R}_{\geq 0}$ by

$$\tau(e) = \begin{cases} e.\mathsf{fields}[\mathsf{timestamp}], & \mathsf{type} \in \{\mathsf{NVTX\_PUSH}, \mathsf{NVTX\_POP}\}, \\ e.\mathsf{fields}[\mathsf{start}], & \mathsf{type} \in \{\mathsf{RUNTIME}, \mathsf{DRIVER}\}, \\ e.\mathsf{fields}[\mathsf{gpu\_start}], & \mathsf{type} = \mathsf{KERNEL}. \end{cases}$$

All subsequent steps sort and compare events only through $\tau$.

**Range reconstruction and semantic partition.** For each process–thread pair $(p, t)$, we first sort the per-thread stream $\mathcal{E}_{p,t} \subset \mathcal{E}$ by $\tau$ and maintain a stack $S_{p,t}$. A push event creates a half-open NVTX range $(\mathsf{id}, \mathsf{name}, \mathsf{start} = \tau)$ that is pushed onto $S_{p,t}$; the matching pop closes the most recent item and yields a completed range

$$r = (\mathsf{id}, \mathsf{name}, \mathsf{pid} = p, \mathsf{tid} = t, \mathsf{start}, \mathsf{end}), \qquad \mathsf{start} < \mathsf{end}.$$

Collecting all completed ranges gives $\mathcal{R}$, and the nesting relation

$$r_1 \prec r_2 \quad :\Longleftrightarrow \quad r_2.\mathsf{start} \leq r_1.\mathsf{start} < r_1.\mathsf{end} \leq r_2.\mathsf{end}$$

induces, for each $(p, t)$, a forest that captures intra-thread hierarchy.

On top of the reconstructed ranges, we apply a rule-based classifier $\Pi : \mathcal{R} \to \{\mathsf{STEP}, \mathsf{FRONTEND}, \mathsf{BACKEND}, \mathsf{COMM}, \mathsf{OTHER}\}$ based on $\mathsf{name}$ patterns to obtain semantic partitions $\mathcal{R}_{\mathsf{STEP}}, \mathcal{R}_{\mathsf{FRONTEND}}, \mathcal{R}_{\mathsf{BACKEND}}, \mathcal{R}_{\mathsf{COMM}}$. Each step $s \in \mathcal{R}_{\mathsf{STEP}}$ carries a request set $\mathcal{Q}(s) \subseteq \mathbb{U}$ parsed from its NVTX message, which is propagated to descendants via the nesting forest unless a stricter subset is encountered.

**Attribution and per-request unification.** Let $\mathcal{A}_{\mathsf{host}} \subset \mathcal{E}$ be host API calls ($\mathsf{RUNTIME} \cup \mathsf{DRIVER}$) and $\mathcal{K} \subset \mathcal{E}$ be device kernels. For a range $r$ with interval $I(r) = [r.\mathsf{start}, r.\mathsf{end}]$, define host attribution

$$\mathsf{host}(r) = \{ a \in \mathcal{A}_{\mathsf{host}} \mid \tau(a) \in I(r) \}.$$

Bind kernels to a host call with the correlation map

$$\Gamma(a) = \{ k \in \mathcal{K} \mid (a.\mathsf{tid}, a.\mathsf{cid}) = (k.\mathsf{tid}, k.\mathsf{cid}) \wedge \tau(a) \leq \tau(k) \},$$

and set $\mathsf{kernels}(r) = \bigcup_{a \in \mathsf{host}(r)} \Gamma(a)$. Let $\mathcal{V} = \mathcal{R}_{\mathsf{STEP}} \cup \mathcal{R}_{\mathsf{FRONTEND}} \cup \mathcal{R}_{\mathsf{BACKEND}}$ and $\mathcal{H} = \mathcal{R}_{\mathsf{COMM}}$. For a step $s$, define

$$\mathcal{T}^v(s) = \{ r \in \mathcal{V} \mid r \preceq s \}, \qquad \mathcal{T}^h(s) = \{ c \in \mathcal{H} \mid c \preceq s \}.$$

For each request $q \in \mathcal{Q}(s)$, project to

$$\mathcal{T}_q^v(s) = \{ r \in \mathcal{T}^v(s) \mid q \in \mathcal{Q}(r) \}, \qquad \mathcal{T}_q^h(s) = \{ c \in \mathcal{T}^h(s) \mid q \in \mathcal{Q}(c) \}.$$

Form the node set $\mathcal{N}_q = \mathcal{T}_q^v(s) \cup \mathcal{T}_q^h(s)$. Edges are given by ancestry inside $\mathcal{T}_q^v(s)$ and by attaching each $c \in \mathcal{T}_q^h(s)$ to the nearest enclosing $r \in \mathcal{T}_q^v(s)$ with $I(c) \subseteq I(r)$. Selecting the nearest ancestor for every node yields an arborescence rooted at $s$, the per-request call-chain tree $\mathcal{C}_q(s)$ with attached host calls and paired kernels.

Each $\mathcal{C}_q(s)$ is serialized to structured JSON and to Chrome trace format, preserving node intervals, categorical tags, host API calls, and paired kernels for visualization and for downstream analysis. The complete pseudocode is provided in Algorithm 1. An example merged call chain is provided in Appendix C.

---

**Algorithm 1** Call-Chain Merging Algorithm

---

1: **Input:** raw events $\mathcal{E}$ with unified time $\tau$
2: **Output:** per-request trees $\{\mathcal{C}_q(s)\}$
3: sort $\mathcal{E}$ by $(\mathsf{pid}, \mathsf{tid}, \tau)$
4: **for** each $(p, t)$ **do**
5:      reconstruct NVTX ranges $\mathcal{R}$ by stack scanning over $\mathcal{E}_{p,t}$
6: **end for**
7: classify $\mathcal{R}$ into $\mathcal{R}_{\mathsf{STEP}}, \mathcal{R}_{\mathsf{FRONTEND}}, \mathcal{R}_{\mathsf{BACKEND}}, \mathcal{R}_{\mathsf{COMM}}$; parse $\mathcal{Q}(s)$ for $s \in \mathcal{R}_{\mathsf{STEP}}$
8: extract host calls $\mathcal{A}_{\mathsf{host}}$ and kernels $\mathcal{K}$; index $\mathcal{A}_{\mathsf{host}}$ by $(\mathsf{tid}, \mathsf{cid})$
9: **for** each $r \in \mathcal{R}$ **do**
10:      $\mathsf{host}(r) \leftarrow \{a \in \mathcal{A}_{\mathsf{host}} \mid \tau(a) \in [r.\mathsf{start}, r.\mathsf{end}]\}$
11:      $\mathsf{kernels}(r) \leftarrow \bigcup_{a \in \mathsf{host}(r)} \Gamma(a)$
12: **end for**
13: **for** each step $s \in \mathcal{R}_{\mathsf{STEP}}$ **do**
14:      $\mathcal{T}^v(s) \leftarrow \{r \in \mathcal{V} \mid r \preceq s\}, \mathcal{T}^h(s) \leftarrow \{c \in \mathcal{H} \mid c \preceq s\}$
15:      **for** each $q \in \mathcal{Q}(s)$ **do**
16:          $\mathcal{T}_q^v(s) \leftarrow \{r \in \mathcal{T}^v(s) \mid q \in \mathcal{Q}(r)\}, \mathcal{T}_q^h(s) \leftarrow \{c \in \mathcal{T}^h(s) \mid q \in \mathcal{Q}(c)\}$
17:          attach each $c \in \mathcal{T}_q^h(s)$ to the nearest enclosing $r \in \mathcal{T}_q^v(s)$
18:          output $\mathcal{C}_q(s)$ as the arborescence rooted at $s$
19:      **end for**
20: **end for**
21: **return** $\{\mathcal{C}_q(s)\}$

---

### 3.3 ANOMALY DETECTION AND ANALYSIS

End-to-end traces expose heterogeneous signals that mix numeric measurements and semantic context. Numeric features such as operator self-time, host and device API durations, kernel counts, data movement, and stream overlap exhibit stable distributions under consistent hardware, software, and workload mix. By the central limit principle and mixture modeling, these marginals are often well approximated by Gaussian components or their mixtures (Rouaud, 2013; Reynolds, 2009). Semantic cues such as operator names, backend tags, kernel families, communication sizes, phase markers, and short logs carry decisive information about root causes that is difficult to encode as fixed thresholds. Prior work has shown that statistical observability complements systems diagnosis (Xu et al., 2025; Qureshi et al., 2023). Large language models can read compact structured text and apply light logical constraints in a way similar to human SRE practice, which allows them to exploit the semantic context that purely numeric detectors ignore. We therefore decompose detection into two stages. A *probabilistic* Stage I provides calibrated numeric confidence for each operator instance, while a *reasoning* Stage II consumes a compact structural summary to yield step-level decisions and operator-level localization, with the former constraining the search space and mitigating hallucination and the latter generalizing to unseen patterns by leveraging textual context.

#### 3.3.1 STATISTICAL MODELING WITH GMM

Let $\mathcal{T}$ be the set of operator families and $u \in \{v, h\}$ the view. For each instance $x$, construct

$$\phi(x) \in \mathbb{R}^d$$

with components for self-time, CUDA runtime/driver time and counts, kernel counts and totals, approximate bytes moved, stream–overlap ratios, and (if applicable) communication size and world size. Apply elementwise $\log(1 + \cdot)$ and per-step or per-family standardization $\phi'(x) = \mathrm{Diag}(\sigma_{u,t})^{-1}(\phi(x) - \mu_{u,t})$, with $(\mu_{u,t}, \sigma_{u,t})$ estimated from normal or robust statistics.

**GMM responsibilities and scores.** Within step $s$, view $u$, family $t$, form $\mathcal{X}_{s,u,t} = \{\phi'(x_i)\}_{i=1}^m \subset \mathbb{R}^d$ and fit a $K$-component mixture

$$p(x) = \sum_{k=1}^K \pi_k \, \mathcal{N}(x \mid \mu_k, \Sigma_k), \qquad \gamma_k(x) = \frac{\pi_k \, \mathcal{N}(x \mid \mu_k, \Sigma_k)}{\sum_{j=1}^K \pi_j \, \mathcal{N}(x \mid \mu_j, \Sigma_j)},$$

where $\Theta = \{\pi_k, \mu_k, \Sigma_k\}_{k=1}^K$ is learned by EM and $K$ is chosen via BIC or a small grid. Let $d_k(x) = \sqrt{(x - \mu_k)^\top \Sigma_k^{-1}(x - \mu_k)}$ and define a central index set $\mathcal{C} \subseteq \{1, \ldots, K\}$ comprised of components with large $\pi_k$ and small $\mathbb{E}[d_k(x)]$. The per-instance confidence and anomaly score are

$$\rho(x) = \sum_{k \in \mathcal{C}} \gamma_k(x), \qquad a(x) = 1 - \rho(x).$$

**Aggregation and stepwise selection.** For step $s$, family $t$, sort confidences $\rho_{(1)} \leq \cdots \leq \rho_{(m)}$. For a quantile $q \in (0, 1]$,

$$\bar{\rho}_{s,t} = \frac{1}{\lfloor qm \rfloor} \sum_{i=1}^{\lfloor qm \rfloor} \rho_{(i)}, \qquad A_{s,t} = 1 - \bar{\rho}_{s,t}.$$

With a robust selector $R$ (e.g., percentile or $\mathrm{median} + \lambda\,\mathrm{MAD}$),

$$\theta_s = R\big(\{A_{s,t}\}_{t \in \mathcal{T}}\big), \qquad \tilde{\mathcal{O}}_s = \{\, t \in \mathcal{T} \mid A_{s,t} \geq \theta_s \,\},$$

retaining the top $M$ by $A_{s,t}$ if $|\tilde{\mathcal{O}}_s| > M$.

Stage I thus fits Gaussian mixtures over operator features to capture multi-modal normal regimes and produces calibrated numeric confidences that both surface suspicious families and constrain the search space for Stage II reasoning. The pseudocode is given in Algorithm 2.

---

**Algorithm 2** GMM Scoring on Per-Step Families

---

1: **Input:** call-chain tree $\mathcal{T}$, feature map $\phi$, views $u \in \{v, h\}$, candidate $K$, quantile $q$, selector $R$, top $M$
2: **Output:** scores $\{A_{s,t}\}$, candidates $\{\tilde{\mathcal{O}}_s\}$
3: **for** each step $s \in \mathcal{T}$ **do**
4:     **for** each $u$ and $t$ **do**
5:         build $\mathcal{X}_{s,u,t} = \{\phi'(x)\}$; fit GMM $\Theta$; compute $\gamma_k(x)$
6:         choose $\mathcal{C}$; set $\rho(x) = \sum_{k \in \mathcal{C}} \gamma_k(x)$
7:         $\bar{\rho}_{s,t} \leftarrow \frac{1}{\lfloor qm \rfloor} \sum_{i=1}^{\lfloor qm \rfloor} \rho_{(i)}$; $A_{s,t} \leftarrow 1 - \bar{\rho}_{s,t}$
8:     **end for**
9:     $\theta_s \leftarrow R(\{A_{s,t}\}_t)$; $\tilde{\mathcal{O}}_s \leftarrow \{t : A_{s,t} \geq \theta_s\}$; keep top $M$
10: **end for**
11: **return** $\{A_{s,t}\}, \{\tilde{\mathcal{O}}_s\}$

---

### 3.3.2 LLM Reasoning with Structural Constraints

For each step $s$ we build a compact summary $\mathsf{J}_s = g(\mathcal{T}_s, \tilde{\mathcal{O}}_s)$ that contains

$$\mathsf{J}_s = \Big( \mathsf{id}(s), \ \{(t, A_{s,t})\}_{t \in \tilde{\mathcal{O}}_s}, \ \mathsf{parents}(t), \ \mathsf{kernels}(t), \ \mathsf{apis}(t), \ \mathsf{context}(t) \Big),$$

where $\mathsf{parents}(t)$ encodes parent–child links in the call-chain tree, and $\mathsf{kernels}(t)$, $\mathsf{apis}(t)$, $\mathsf{context}(t)$ provide minimal semantic tags.

**Decision function.** Let $f_\mathcal{P}$ be the LLM queried with a fixed prompt $\mathcal{P}$. Deterministic decoding and a strict output schema $\mathcal{S}$ are enforced by a projection $\Pi_\mathcal{S}$.

$$(\hat{y}_s, \hat{\mathcal{O}}_s) = \Pi_\mathcal{S}\big(f_\mathcal{P}(\mathsf{J}_s)\big), \qquad \hat{y}_s \in \{\mathsf{normal}, \mathsf{abnormal}\}.$$

**Fusion rule.** The final step label and faulty set are

$$\mathsf{label}(s) = \begin{cases} \mathsf{abnormal}, & \exists t \in \tilde{\mathcal{O}}_s \cap \hat{\mathcal{O}}_s, \\ \mathsf{normal}, & \text{otherwise}, \end{cases} \qquad \mathcal{O}_s = \tilde{\mathcal{O}}_s \cap \hat{\mathcal{O}}_s.$$

This rule respects probabilistic evidence from GMM and requires LLM confirmation, which stabilizes decisions and reduces false positives.

Stage II utilizes the calibrated confidences from Stage I together with structural and semantic summaries to perform context-aware reasoning. By combining operator candidates, step-level structure, and auxiliary textual cues, the LLM decides whether a step is abnormal and identifies faulty operators in a rule-guided manner. In this division of labor, Stage I bounds the decision region numerically, while Stage II generalizes beyond distributional regularities and leverages context, jointly improving robustness and interpretability. The complete pseudocode of Stage II is provided in Algorithm 3. Detailed prompts used for the LLM are provided in Appendix D.

---

**Algorithm 3** LLM Decision with Schema Enforcement

---

1: **Input:** per-step tree $\mathcal{T}_s$, candidates $\tilde{\mathcal{O}}_s$, scores $\{A_{s,t}\}$, prompt $\mathcal{P}$, schema $\mathcal{S}$
2: **Output:** label$(s)$, $\mathcal{O}_s$
3: build $\mathsf{J}_s = g(\mathcal{T}_s, \tilde{\mathcal{O}}_s)$
4: $(\hat{y}_s, \hat{\mathcal{O}}_s) \leftarrow \Pi_{\mathcal{S}}\big(f_{\mathcal{P}}(\mathsf{J}_s)\big)$
5: $\mathcal{O}_s \leftarrow \tilde{\mathcal{O}}_s \cap \hat{\mathcal{O}}_s$
6: label$(s) \leftarrow$ abnormal if $\mathcal{O}_s \neq \emptyset$ else normal
7: **return** label$(s)$, $\mathcal{O}_s$

---

## 4 EXPERIMENTS

### 4.1 EXPERIMENTAL SETUP

We evaluate TRUFFLD on a dual-node GPU cluster deployed with `vLLM` serving `Qwen3-8B`. Each node is equipped with an Intel Xeon Gold 6326 CPU @ 2.90 GHz, 128 GB RAM, three NVIDIA A40 (48 GB) GPUs, and a ConnectX-6 NIC, giving a total of six GPUs across the cluster. Both online and offline workloads are used: the online load replays production-like request traces with realistic batching and concurrency, while the offline load reuses the same model and prompts under controlled replay to ensure repeatability.

### 4.2 DATASET CONSTRUCTION

We collect per-request call-chain traces under vertical (within-node) and horizontal (cross-node communication) views, covering 100 tasks (50 online, 50 offline) and 200 process IDs. In total we curate 3,264 labeled step-level traces (1,632 per view). Table 1 summarizes step outcomes together with operator-level counts and error rates. Detailed procedures for dataset construction, including the fault-injection methodology at software, CUDA, hardware, and network layers, are provided in Appendix E.

Table 1: Overview of Step- and Operator-Level Trace Statistics.

| View | Steps | Step errors | Step error rate | Ops | Op errors | Op error rate |
|---|---|---|---|---|---|---|
| Vertical | 1,632 | 817 | 50.06% | 4,963,907 | 899,404 | 18.12% |
| Horizontal | 1,632 | 805 | 49.33% | 126,119 | 21,706 | 17.21% |
| Overall | 3,264 | 1,622 | 49.69% | 5,090,026 | 921,110 | 18.10% |

### 4.3 RESULTS

We compare TRUFFLD with a diverse set of unsupervised and supervised baselines commonly used in anomaly detection: classical clustering methods (KMeans, DBSCAN, Isolation Forest, GMM) (MacQueen, 1967; Ester et al., 1996; Liu et al., 2008; Reynolds, 2009), supervised learners (SVM, Random Forest, XGBoost) (Steinwart & Christmann, 2008; Breiman, 2001; Chen & Guestrin, 2016), and recent log-based or deep anomaly detection methods including Robustlog, LAnoBERT, Logs2Graphs, and MAD-GAN (Zhang et al., 2019; Lee et al., 2021; Li et al., 2023; 2019). At the step level, we report Accuracy, Precision, and F1. At the operator level, we report (i) **Macro** metrics (Precision, F1, Jaccard) across operator types, and (ii) **Macro$_+$** metrics computed

only over operator types with positive support (i.e., at least one true error). Macro$_+$ emphasizes behavior when faults actually occur, which is critical for incident response. The results are summarized in Table 2. Detailed baseline configurations and the formal definition of all metrics appear in Appendix F, and representative error cases are illustrated in Appendix G.

Table 2: Comparison of end-to-end detection methods on horizontal (cross-node) and vertical (within-node) views.

| Method | Step | | | Operator Macro | | | Operator Macro$_+$ | | |
|---|---|---|---|---|---|---|---|---|---|
| | Accuracy | Precision | F1 | Precision | F1 | Jaccard | Precision$_+$ | F1$_+$ | Jaccard$_+$ |
| *Horizontal view* | | | | | | | | | |
| TRUFFLD (ours) | **0.926** | **0.961** | **0.922** | **0.435** | **0.880** | **0.842** | **0.898** | **0.789** | **0.711** |
| Robustlog | 0.874 | 0.735 | 0.771 | 0.267 | 0.384 | 0.238 | 0.566 | 0.597 | 0.425 |
| LAnoBERT | 0.882 | 0.822 | 0.783 | 0.259 | 0.381 | 0.236 | 0.649 | 0.672 | 0.507 |
| Logs2Graphs | 0.825 | 0.603 | 0.684 | 0.254 | 0.383 | 0.237 | 0.514 | 0.596 | 0.424 |
| MAD-GAN | 0.819 | 0.636 | 0.706 | 0.246 | 0.382 | 0.236 | 0.460 | 0.579 | 0.407 |
| KMeans | 0.457 | 0.455 | 0.611 | 0.176 | 0.244 | 0.176 | 0.386 | 0.533 | 0.386 |
| DBSCAN | 0.457 | 0.457 | 0.628 | 0.187 | 0.259 | 0.187 | 0.408 | 0.567 | 0.408 |
| IsolationForest | 0.433 | 0.444 | 0.604 | 0.175 | 0.244 | 0.175 | 0.383 | 0.533 | 0.383 |
| GMM | 0.494 | 0.470 | 0.603 | 0.163 | 0.223 | 0.163 | 0.355 | 0.488 | 0.355 |
| SVM | 0.463 | 0.373 | 0.302 | 0.037 | 0.056 | 0.037 | 0.082 | 0.123 | 0.082 |
| RandomForest | 0.805 | 0.713 | 0.818 | 0.178 | 0.248 | 0.178 | 0.388 | 0.543 | 0.388 |
| XGBoost | 0.817 | 0.817 | 0.795 | 0.150 | 0.207 | 0.150 | 0.328 | 0.452 | 0.328 |
| *Vertical view* | | | | | | | | | |
| TRUFFLD (ours) | 0.893 | **1.000** | **0.881** | **0.392** | **0.727** | **0.662** | **0.793** | 0.448 | 0.315 |
| Robustlog | 0.873 | 0.725 | 0.703 | 0.310 | 0.447 | 0.288 | 0.454 | 0.559 | 0.388 |
| LAnoBERT | 0.875 | 0.675 | 0.726 | 0.214 | 0.340 | 0.205 | 0.442 | **0.572** | **0.401** |
| Logs2Graphs | **0.922** | 0.852 | 0.791 | 0.233 | 0.356 | 0.216 | 0.434 | 0.560 | 0.389 |
| MAD-GAN | 0.837 | 0.686 | 0.741 | 0.262 | 0.392 | 0.244 | 0.425 | 0.558 | 0.387 |
| KMeans | 0.518 | 0.518 | 0.683 | 0.191 | 0.278 | 0.191 | 0.369 | 0.537 | 0.369 |
| DBSCAN | 0.518 | 0.518 | 0.683 | 0.191 | 0.278 | 0.191 | 0.369 | 0.537 | 0.369 |
| IsolationForest | 0.537 | 0.528 | 0.691 | 0.191 | 0.278 | 0.191 | 0.369 | 0.537 | 0.369 |
| GMM | 0.549 | 0.545 | 0.644 | 0.149 | 0.219 | 0.149 | 0.288 | 0.422 | 0.288 |
| SVM | 0.701 | 0.634 | 0.776 | 0.191 | 0.278 | 0.191 | 0.369 | 0.537 | 0.369 |
| RandomForest | 0.854 | 0.942 | 0.844 | 0.145 | 0.212 | 0.145 | 0.280 | 0.409 | 0.280 |
| XGBoost | 0.780 | 0.930 | 0.746 | 0.119 | 0.174 | 0.119 | 0.230 | 0.336 | 0.230 |

Overall, TRUFFLD consistently achieves the strongest step-level accuracy and F1 on both views while maintaining near-perfect precision. Macro$_+$ further shows that TRUFFLD remains competitive when conditioning on fault-present classes, which is the regime most critical to incident response. While some baselines achieve isolated strengths, TRUFFLD delivers balanced and robust improvements across both views and both evaluation levels, demonstrating its overall effectiveness for full-stack anomaly detection.

## 5 CONCLUSION

We presented TRUFFLD, a non-intrusive framework that bridges low-level tracing and fine-grained cross-layer representations for LLM inference diagnosis. The system profiles execution from vertical and horizontal perspectives using NVTX and CUPTI, and merges multi-source events into per-request call-chain trees with a unified time base. Building on this representation, a two-stage analysis combines Gaussian Mixture Models for calibrated numeric confidences with an LLM-driven reasoning stage for step-level decisions and operator-level localization. Experiments on a multi-node cluster serving Qwen-8B with online and offline workloads demonstrate near-perfect step-level detection, state-of-the-art operator-level accuracy, and low deployment overhead.[2]

---

[2]Detailed measurements of overhead are provided in Appendix H.

ETHICS STATEMENT

All authors have read and agreed to adhere to the ICLR Code of Ethics[3]. This work does not involve human subjects, personally identifiable information, or sensitive demographic data. Our study focuses on system-level tracing and anomaly detection for distributed LLM inference, and therefore does not raise fairness, discrimination, or security risks beyond standard considerations in system monitoring. All experiments were conducted on controlled compute clusters with synthetic or open-source workloads, ensuring no violation of privacy or legal compliance.

REPRODUCIBILITY STATEMENT

We have made significant efforts to ensure reproducibility. The proposed methods are described in detail in Sections 3.2.1–3.3.2, with complete algorithms given in Algorithms 1, 2, and 3. Baseline configurations and metric definitions are provided in Appendix F, and representative error cases in Appendix G. An anonymized code package and scripts for data preprocessing, model training, and evaluation are included in the supplemental materials. These materials collectively ensure that the results in this paper can be independently verified.

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

# A   PYTORCH NVTX INJECTION TABLE

We list representative entries of the NVTX injection table used to instrument PyTorch front-end functions.

```
PyTorch NVTX Injection Table (excerpt)

{
    ...
    "domain": "PyTorch",
    "module": "torch",
    "functions": ["Tensor.to"]
},
{
    "domain": "PyTorch",
    "module": "torch.utils.data.dataloader",
    "functions": ["_BaseDataLoaderIter.__next__"]
},
{
    "domain": "PyTorch",
    "module": "torch.autograd",
    "functions": ["backward"]
},
{
    "domain": "PyTorch",
    "module": "torch.nn.functional",
    "functions": [
        "conv1d", "conv2d", "conv3d",
        "conv_transpose1d", "conv_transpose2d", "conv_transpose3d",
        "avg_pool1d", "avg_pool2d", "avg_pool3d",
        "fractional_max_pool2d", "fractional_max_pool3d",
        "max_pool1d", "max_pool2d_with_indices",
        "adaptive_max_pool1d", "adaptive_max_pool2d", ...
    ]
},
{
    "domain": "PyTorch",
    "module": "torch.optim",
    "functions": [
        "Adafactor.step", "Adadelta.step", "Adagrad.step", "Adam.step",
        ↪  "Adamax.step", "AdamW.step", "ASGD.step", "LBFGS.step", "NAdam.step",
        ↪  "Optimizer.step", "RAdam.step", "RMSprop.step", "Rprop.step",
        ↪  "SGD.step", "SparseAdam.step", ...
    ]
},
{
    "domain":"PyTorch",
    "module": "torch.nn.modules",
    "functions": [
        "Module._call_impl", "Linear.forward", "GELU.forward", "ReLU.forward",
        ↪  "Bilinear.forward", "Identity.forward", "LazyLinear.forward",
        ↪  "Linear.forward", "CELU.forward", "ELU.forward", "GELU.forward",
        ↪  "GLU.forward", "Hardshrink.forward", ...
    ]
    ...
}
```

*Note.* Due to space limitations, we only show a subset of the full list. The complete injection table is included in the supplemental materials.

## B  EXAMPLE OF COLLECTED RAW EVENTS

We report representative raw events collected from NVTX and CUPTI. Each record includes the event type, name or template, nanosecond timestamps, correlation identifiers, and process or thread identifiers.

**Raw events (excerpt)**

```
...
{ "type": "RUNTIME", "cbid": 165, "name": "cudaDeviceSynchronize_v3020", "start":
↪ 1758730718542343309, "end": 1758730718542353355, "duration": 10046,
↪ "correlation_id": 255, "process_id": 1603384, "thread_id": 2034481024 }
{ "type": "NVTX_MARKER", "name": "aten::arange, op_id = 1", "timestamp":
↪ 1758730719435661224, "id": 1, "process_id": 1603384, "thread_id": 2034481024 }
{ "type": "NVTX_MARKER", "name": "aten::empty, op_id = 2", "timestamp":
↪ 1758730719435679663, "id": 2, "process_id": 1603384, "thread_id": 2034481024 }
{ "type": "NVTX_MARKER", "name": "null", "timestamp": 1758730719435700922, "id": 2,
↪ "process_id": 1603384, "thread_id": 2034481024 }
{ "type": "NVTX_MARKER", "name": "aten::arange, op_id = 3", "timestamp":
↪ 1758730719435707885, "id": 3, "process_id": 1603384, "thread_id": 2034481024 }
{ "type": "NVTX_MARKER", "name": "aten::resize_, op_id = 4", "timestamp":
↪ 1758730719435733082, "id": 4, "process_id": 1603384, "thread_id": 2034481024 }
{ "type": "NVTX_MARKER", "name": "null", "timestamp": 1758730719435741672, "id": 4,
↪ "process_id": 1603384, "thread_id": 2034481024 }
{ "type": "NVTX_MARKER", "name": "null", "timestamp": 1758730719435797736, "id": 3,
↪ "process_id": 1603384, "thread_id": 2034481024 }
{ "type": "NVTX_MARKER", "name": "null", "timestamp": 1758730719435798139, "id": 1,
↪ "process_id": 1603384, "thread_id": 2034481024 }
{ "type": "NVTX_MARKER", "name": "aten::reshape, seq = 0, op_id = 5", "timestamp":
↪ 1758730719435848288, "id": 5, "process_id": 1603384, "thread_id": 2034481024 }
{ "type": "NVTX_MARKER", "name": "aten::view, seq = 0, op_id = 6", "timestamp":
↪ 1758730719435854942, "id": 6, "process_id": 1603384, "thread_id": 2034481024 }
{ "type": "NVTX_MARKER", "name": "null", "timestamp": 1758730719435882099, "id": 6,
↪ "process_id": 1603384, "thread_id": 2034481024 }
{ "type": "NVTX_MARKER", "name": "null", "timestamp": 1758730719435882531, "id": 5,
↪ "process_id": 1603384, "thread_id": 2034481024 }
{ "type": "NVTX_MARKER", "name": "aten::view, seq = 0, op_id = 7", "timestamp":
↪ 1758730719435894048, "id": 7, "process_id": 1603384, "thread_id": 2034481024 }
{ "type": "NVTX_MARKER", "name": "null", "timestamp": 1758730719435903986, "id": 7,
↪ "process_id": 1603384, "thread_id": 2034481024 }
{ "type": "NVTX_MARKER", "name": "aten::unbind, seq = 0, op_id = 8", "timestamp":
↪ 1758730719435920162, "id": 8, "process_id": 1603384, "thread_id": 2034481024 }
{ "type": "NVTX_MARKER", "name": "aten::select, op_id = 9", "timestamp":
↪ 1758730719435943449, "id": 9, "process_id": 1603384, "thread_id": 2034481024 }
{ "type": "NVTX_MARKER", "name": "aten::as_strided, op_id = 10", "timestamp":
↪ 1758730719435957904, "id": 10, "process_id": 1603384, "thread_id": 2034481024 }
{ "type": "NVTX_MARKER", "name": "null", "timestamp": 1758730719435961639, "id": 10,
↪ "process_id": 1603384, "thread_id": 2034481024 }
{ "type": "NVTX_MARKER", "name": "null", "timestamp": 1758730719435963982, "id": 9,
↪ "process_id": 1603384, "thread_id": 2034481024 }
{ "type": "RUNTIME", "cbid": 41, "name": "cudaMemcpyAsync_v3020", "start":
↪ 1758730719803400971, "end": 1758730719803404014, "duration": 3043,
↪ "correlation_id": 1516, "process_id": 1603384, "thread_id": 2034481024 }
{ "type": "RUNTIME", "cbid": 131, "name": "cudaStreamSynchronize_v3020", "start":
↪ 1758730719803404207, "end": 1758730719803409531, "duration": 5324,
↪ "correlation_id": 1517, "process_id": 1603384, "thread_id": 2034481024 }
{ "type": "NVTX_MARKER", "name": "null", "timestamp": 1758730719803410478, "id": 435,
↪ "process_id": 1603384, "thread_id": 2034481024 }
{ "type": "NVTX_MARKER", "name": "null", "timestamp": 1758730719803410799, "id": 433,
↪ "process_id": 1603384, "thread_id": 2034481024 }
{ "type": "KERNEL", "name":
↪ "_ZN2at6native29vectorized_elementwise_kernelILi4ENS0_13AUnaryFunctorIfffNS0",
↪ "gpu_start": 1758730719782552356, "gpu_end": 1758730719782557188, "duration": 4832,
↪ "correlation_id": 1390 }
...
```

*Note.* The full JSON is included in the supplemental materials, and the excerpt above contains a randomly sampled subset due to space limits.

## C  EXAMPLE OF MERGED CALL CHAIN

We provide a representative excerpt of one merged per-request call-chain tree $\mathcal{C}_q(s)$. The structure preserves the step root, selected front-end/back-end operators, associated host runtime calls, and paired GPU kernels, with timing in nanoseconds.

```
Call-chain tree (excerpt)

{
  "name": "step_execute_model[request_ids:0_1_2]",
  "start": 0,
  "end": 9353660367,
  "layer": "step_execute_model",
  "is_error": true,
  "children": [
    {
      "name": "torch.nn.functional.embedding .../torch/nn/functional.py:2437",
      "start": 3389877,
      "end": 3529065,
      "layer": "torch_frontend_op",
      "is_error": true,
      "children": [
        { "name": "aten::empty, op_id = 426373", "start": 3404516, "end": 3408130,
          "layer": "torch_backend_op", "is_error": false, "children": [] },
        { "name": "aten::resize_, op_id = 426374", "start": 3413076, "end": 3417470,
          "layer": "torch_backend_op", "is_error": false, "children": [] },
        { "name": "aten::index_select, op_id = 426372", "start": 3417570, "end":
        ↪  3486452,
          "layer": "torch_backend_op", "is_error": true,
          "children": [
            { "name": "cudaLaunchKernel_v7000", "start": 3442283, "end": 3460849,
              "layer": "runtime_call", "is_error": false,
              "children": [
                { "name": "_ZN2at6native...indexSelectLargeIndex...", "start": 3450318,
                  "end": 3460749, "layer": "kernel", "is_error": false, "children": []
                  ↪  }
              ] }
          ] },
        { "name": "aten::embedding, op_id = 426371", "start": 3486552, "end": 3527296,
          "layer": "torch_backend_op", "is_error": false, "children": [] }
      ]
    },
    ...
    {
      "name": "torch.nn.functional.linear at :0",
      "start": 4686570, "end": 5327350, "layer": "torch_frontend_op", "is_error": true,
      "children": [
        { "name": "aten::transpose, op_id = 426389", "start": 4701977, "end": 4706278,
          "layer": "torch_backend_op", "is_error": false, "children": [] },
        { "name": "aten::mm, op_id = 426392", "start": 4721730, "end": 5064790,
          "layer": "torch_backend_op", "is_error": true,
          "children": [
            { "name": "cudaMemsetAsync_v3020", "start": 4802165, "end": 4815811,
              "layer": "runtime_call", "is_error": false, "children": [] },
            { "name": "cudaLaunchKernel_v7000", "start": 4818552, "end": 5003238,
              "layer": "runtime_call", "is_error": false,
              "children": [
                { "name": "turing_fp16_s1688gemm_fp16_128x64_sliced1x2...", "start":
                ↪  4829734,
                  "end": 5003138, "layer": "kernel", "is_error": false, "children": []
                  ↪  }
              ] }
          ] },
        { "name": "aten::linear, op_id = 426387", "start": 5186223, "end": 5323477,
          "layer": "torch_backend_op", "is_error": false, "children": [] }
      ]
    }
  ]
}
```

*Note.* The full JSON for this call-chain is included in the supplemental materials, and we show only a compact excerpt above due to space limits.

# D  LLM PROMPT TEMPLATE

The exact instruction provided to the LLM in 3.3.2 is shown below.

We use `gpt-5-nano-2025-08-07` as the backend model.

## LLM Prompt

```
You are a performance diagnostics expert for distributed LLM inference traces.

You will receive a compact feature object for ONE step:
{
  "kind": "vertical" | "horizontal",
  "ops_count": <int>,
  "ops_prob": [
    { "name": <string>, "prob_center": <float in [0,1]>, "parent": <string or null> },
    ...
  ]
}

Semantics:
- prob_center is the probability that an operator instance belongs to the NORMAL
↪  center-cluster set (one or multiple centers).
  Smaller prob_center => more abnormal (farther from all centers). Larger => more
  ↪  normal.
- parent (if present) is the most frequent direct parent for that operator name in THIS
↪  STEP.
- Errors propagate UPWARD ONLY (child -> parent), not downward.

Definitions:
- Strong anomaly: prob_center is extremely small, typically shown in scientific
↪  notation (e.g., 1e-6, 3.2e-12).
  For numeric comparison, treat prob_center < 1e-6 as STRONG.
- Weak anomaly: 0 < prob_center < 0.5 and NOT strong.

Your tasks:
1) Decide if this step is anomalous (true/false).
2) List operator NAMES that are anomalous (subset of names that appear in ops_prob).

Decision policy:
1. Compute S = number of operators classified as STRONG anomalies in this step.
2. Step-level rule (different by kind):
     - If kind == "horizontal": if S <= 1, set step_anomaly=false and op_error_names=[].
     - If kind == "vertical"  : if S / ops_count < 0.10, set step_anomaly=false and
     ↪  op_error_names=[].
3. Otherwise (i.e., enough STRONG anomalies):
     - Set step_anomaly=true.
     - Include ALL STRONG anomalies in op_error_names.
     - Optionally include WEAK anomalies that appear meaningfully related (e.g., share a
     ↪  parent with strong anomalies or form a coherent group of low probabilities).
4. UPWARD PROPAGATION (MANDATORY): For every operator you include, if it has a parent
↪  (non-null) and that parent name exists in this step, ALSO include that parent
↪  (unconditionally). Do not propagate downward.
5. De-duplicate names in the final list. Only include operator names that actually
↪  appear in ops_prob.

Constraints:
- If step_anomaly=false, you MUST return an empty operator list: "op_error_names": [].
- Output must be JSON only, with this exact schema:
{
  "step_anomaly": true|false,
  "op_error_names": ["op_name_1", "op_name_2", ...]
}
```

# E    DATASET CONSTRUCTION AND FAULT INJECTION

In this section, we describe how the evaluation dataset is constructed, with a focus on controlled fault injection across software, CUDA, hardware, and communication layers. Our goal is to create realistic anomalies that stress both the vertical (per-request stack) and horizontal (cross-node communication) views captured by TRUFFLD, while preserving clean baselines for comparison.

**Workload and Baseline Traces.**    We run multi-node, multi-GPU LLM inference under `vLLM` with `Qwen-8B`, generating paired online-style and offline-style request traces. For each run, we first record a fault-free baseline to establish the nominal distributions of operator self-time, CUDA API and kernel durations, data movement, and communication sizes. All experiments use fixed random seeds and pinned software versions to ensure repeatability.

**Software-Layer Fault Injection.**    To emulate compute-side slowdowns in high-level operators, we inject software faults in PyTorch using `pytorchfi` (PyTorch, 2025b). We target frequently exercised families such as matrix multiplication, linear layers, normalization, and activation functions. Each injection stochastically perturbs a subset of calls within a bounded window by (i) delaying the operator body through calibrated sleeps or synthetic work, or (ii) perturbing tensor shapes within valid bounds so that downstream kernels take slower execution paths. Intensities are sampled from a discrete set of delay budgets that preserve functional correctness. Each injected occurrence is tagged with the operator name, callsite, and injection ID so that ground-truth labels align with both vertical chains and operator families.

**CUDA-Layer Fault Injection.**    To emulate device-side anomalies, we introduce CUDA-level perturbations with `DCGM` (NVIDIA, 2025a). We instrument kernels and memory operations that dominate step latency and apply: (i) controlled kernel slowdowns that mimic backpressure or suboptimal scheduling, (ii) transient memory pressure that inflates allocation and copy times, and (iii) brief device throttling events that increase API and kernel latency but keep runs stable. Each event is bounded in duration and frequency to avoid crashes and to resemble intermittent degradations. CUPTI correlation identifiers then bind perturbed kernels to their launching host calls, enabling precise attribution on the merged call-chain.

**Hardware/Resource-Contention Scenarios.**    To simulate contention without altering application code, we co-schedule auxiliary jobs that share GPUs with the primary inference job. These jobs are configured to occupy SMs and memory bandwidth in controlled cycles, producing measurable fluctuations in utilization, memory throughput, temperature, and power. The schedule alternates quiet and busy phases to create realistic performance variance under multi-tenant settings. We record the contention timeline and map it to step intervals so that ground truth is available at both step and operator levels.

**Communication-Layer Fault Injection.**    To exercise horizontal analysis, we perturb inter-node links using `chaosblade` (Chaosblade, 2024). We apply bounded additional latency and packet loss on selected links and windows while keeping connectivity intact. The treatment targets phases with collective communication (e.g., all-reduce, all-gather) so that the impact appears as elongated NCCL operators and shifted overlap with compute. Each perturbation is logged with link endpoints, start and end time, and intensity, enabling consistent alignment to communication operators in the horizontal view.

**Labeling Protocol.**    Each run yields a time-stamped ledger of injections:

$$\mathcal{L} = \{(layer, target, start, end, intensity, id)\}.$$

During post-processing, we intersect $\mathcal{L}$ with the merged call-chain trees. A step is labeled abnormal if any of its enclosed operators or communication segments overlaps a ledger entry with sufficient intensity; otherwise it is normal. At the operator level, labels attach to the smallest enclosing unit (backend operator or NCCL operator) whose interval overlaps the injection window. This yields consistent step-level and operator-level ground truth across vertical and horizontal views.

**Coverage and Balance.**    We distribute injections across layers and intensities to avoid class imbalance and to encourage diversity of root causes: software-layer slowdowns are allocated to common compute operators, CUDA-layer perturbations to kernels and memory paths that are prevalent in decoding, resource contention events to a subset of GPUs per node, and communication perturbations to selected collectives and ranks. Normal runs are interleaved to maintain a near-balanced mix at the step level.

**Limitations.**    While our injections approximate realistic degradations reported in prior work (Xu et al., 2025), they do not cover permanent hardware failures or long-duration outages. Extending to additional failure modes such as PCIe errors or persistent ECC storms is left for future work.

# F BASELINES AND METRICS

In this section we formalize the baselines and evaluation metrics used in §4.3, and we provide detailed configurations for each method.

**Notation.** Let $\mathcal{S}$ be the set of evaluated steps, $|\mathcal{S}| = N$. Each step $s \in \mathcal{S}$ has a binary ground truth $y_s \in \{0, 1\}$ where 1 denotes abnormal. Let $\mathcal{T}$ be the set of operator families. For each $t \in \mathcal{T}$ and step $s$, let $y_{s,t} \in \{0, 1\}$ indicate whether family $t$ contains at least one true faulty operator in step $s$. A method outputs $\hat{y}_s \in \{0, 1\}$ and $\hat{y}_{s,t} \in \{0, 1\}$ at step and operator-family levels, respectively. All methods operate on the same feature map $\phi(\cdot)$ defined in the main paper, with $\log(1 + \cdot)$ transforms and per-scope standardization.

## F.1 UNSUPERVISED BASELINES

**KMeans** (MacQueen, 1967). Given $K \in \{2, 3, 4, 5\}$, fit centroids $\{c_k\}_{k=1}^K$ by minimizing within-cluster $\ell_2$ distortion on training features $\{\phi(x_i)\}_{i=1}^n$. The anomaly score is the distance to the nearest centroid

$$a(x) = \min_k \|\phi(x) - c_k\|_2.$$

We evaluate cosine distance as an ablation but report Euclidean due to better stability under per-step normalization. A decision threshold $\tau$ is chosen on validation data to maximize F1. Predictions follow $\hat{y} = \mathbb{I}\{a(x) \geq \tau\}$ at the appropriate level.

**DBSCAN** (Ester et al., 1996). We use the density-based algorithm with parameters $\varepsilon \in \{0.3, 0.5, 0.7\}$ and minPts $\in \{5, 10, 20\}$. Points labeled *noise* are treated as anomalies with $a(x) = 1$ and clustered points receive $a(x) = 0$. Since DBSCAN is non-parametric and can mark most points as noise under high dimensionality, we precede it with PCA to $d' \in \{8, 16, 32\}$ principal components fitted on the training split.

**Isolation Forest** (Liu et al., 2008). We train an ensemble of random isolation trees with number of estimators $n_{\text{trees}} \in \{100, 200\}$ and subsampling size $\psi \in \{256, 512, 1024\}$. The path length $\ell(x)$ averaged over trees yields the isolation score

$$a(x) = 1 - \exp\left(-\frac{\ell(x)}{c(\psi)}\right),$$

where $c(\psi)$ is the harmonic number based normalization. Larger $a(x)$ implies more anomalous behavior. Threshold $\tau$ is selected on validation to maximize F1.

**GMM (w/o LLM)** (Reynolds, 2009). We fit a Gaussian Mixture Model with components $K \in \{2, 3, 4\}$ and full or diagonal covariance, selected by BIC. The density

$$p(x) = \sum_{k=1}^K \pi_k \mathcal{N}\big(\phi(x) \mid \mu_k, \Sigma_k\big)$$

is estimated by EM. The anomaly score is the negative log-likelihood

$$a(x) = -\log p(x).$$

## F.2 SUPERVISED BASELINES

**SVM** (Steinwart & Christmann, 2008). We consider linear and RBF kernels. For the RBF kernel the grid is $C \in \{0.1, 1, 10\}$ and $\gamma \in \{\frac{1}{d}, \frac{2}{d}, \frac{4}{d}\}$ where $d$ is the feature dimension. Probabilities $\hat{p}(x)$ are obtained via Platt scaling on the validation split.

**Random Forest** (Breiman, 2001). We use $n_{\text{trees}} \in \{200, 400\}$ with maximum depth $\in \{10, 20, \text{None}\}$ and minimum samples per leaf $\in \{1, 2, 4\}$. Class weights are set to "balanced". Probabilities are the empirical class frequency across trees for each sample.

**XGBoost** (Chen & Guestrin, 2016). We train gradient-boosted trees with binary logistic loss. The grid is $n_{\text{trees}} \in \{300, 600\}$, learning rate $\eta \in \{0.05, 0.1\}$, maximum depth $\in \{6, 8\}$, subsample $\in \{0.8, 1.0\}$, and column sample by tree $\in \{0.8, 1.0\}$. Early stopping on validation F1 with patience 20 rounds is used. The output $\hat{p}(x)$ is the sigmoid of the accumulated margin.

### F.3 LOG-BASED AND SEQUENCE BASELINES

We additionally evaluate log-based and sequence-oriented baselines that have been widely used for system anomaly detection. All methods operate on parsed NVTX/CUPTI traces, where raw events are converted into textual templates using the Drain parser. Unless otherwise specified, anomaly scores are first computed on log windows and then aggregated by taking the maximum score within each step and operator family. Thresholds are tuned on validation data to maximize F1.

**Robustlog.** Robustlog (Zhang et al., 2019) extends the DeepLog approach by improving robustness to unstable log tokens. An LSTM is trained to predict the next log template given a sequence of the previous $L$ templates. We search hidden sizes $\{64, 128, 256\}$, number of layers $\{1, 2\}$, dropout $\{0, 0.3\}$, and window lengths $\{20, 50, 100\}$. Anomaly scores are defined as one minus the predicted probability of the actual next template.

**LAnoBERT.** LAnoBERT (Lee et al., 2021) formulates log anomaly detection as masked language modeling. A BERT encoder is fine-tuned on template sequences with random masking of 15% of tokens. We use BERT-base with maximum sequence length 128, batch sizes $\{64, 128\}$, learning rates $\{2 \times 10^{-5}, 5 \times 10^{-5}\}$, and up to 10 epochs of training. Anomaly scores are the average negative log-likelihood of masked tokens.

**Logs2Graphs.** Logs2Graphs (Li et al., 2023) represents each step as a graph of co-occurring templates. Nodes correspond to template IDs, and an edge is added when two templates appear within a window of size $\{10, 20\}$. We employ a two-layer GCN autoencoder with hidden dimensions $\{64, 128\}$. Anomaly scores are based on reconstruction error of the adjacency matrix, and step-level scores are taken as the maximum node score.

**MAD-GAN.** MAD-GAN (Li et al., 2019) applies a generative adversarial network to multivariate time series anomaly detection. We treat operator feature sequences as multivariate series after log-transformation and z-score standardization. The generator and discriminator are implemented with LSTM layers of size $\{64, 128\}$ and window length $\{50, 100\}$. The anomaly score for each window is a weighted sum of reconstruction error and discriminator feature-matching loss.

### F.4 STEP-LEVEL METRICS

Let TP, FP, TN, FN be counts over $\{\hat{y}_s\}$ and $\{y_s\}$. We report

$$\text{Accuracy} = \frac{\text{TP} + \text{TN}}{N}, \quad \text{Precision} = \frac{\text{TP}}{\text{TP} + \text{FP}}, \quad \text{F1} = \frac{2\,\text{TP}}{2\,\text{TP} + \text{FP} + \text{FN}}.$$

### F.5. OPERATOR-LEVEL METRICS

For each $t \in \mathcal{T}$ define

$$\text{TP}_t = \sum_s \mathbb{I}\{\hat{y}_{s,t} = 1, y_{s,t} = 1\}, \text{FP}_t = \sum_s \mathbb{I}\{\hat{y}_{s,t} = 1, y_{s,t} = 0\}, \text{FN}_t = \sum_s \mathbb{I}\{\hat{y}_{s,t} = 0, y_{s,t} = 1\}.$$

Per-family scores are

$$P_t = \frac{\text{TP}_t}{\text{TP}_t + \text{FP}_t + \varepsilon}, \quad R_t = \frac{\text{TP}_t}{\text{TP}_t + \text{FN}_t + \varepsilon}, \quad F1_t = \frac{2P_t R_t}{P_t + R_t + \varepsilon}, \quad J_t = \frac{\text{TP}_t}{\text{TP}_t + \text{FP}_t + \text{FN}_t + \varepsilon}.$$

Macro averages over all families are

$$\text{Macro-}P = \frac{1}{|\mathcal{T}|} \sum_{t \in \mathcal{T}} P_t, \quad \text{Macro-}F1 = \frac{1}{|\mathcal{T}|} \sum_{t \in \mathcal{T}} F1_t, \quad \text{Macro-}J = \frac{1}{|\mathcal{T}|} \sum_{t \in \mathcal{T}} J_t.$$

Let $\mathcal{T}_+ = \{t \in \mathcal{T} : \sum_s y_{s,t} > 0\}$. The positive-support macro averages are

$$\text{Macro}_+\text{-}P = \frac{1}{|\mathcal{T}_+|} \sum_{t \in \mathcal{T}_+} P_t, \quad \text{Macro}_+\text{-}F1 = \frac{1}{|\mathcal{T}_+|} \sum_{t \in \mathcal{T}_+} F1_t, \quad \text{Macro}_+\text{-}J = \frac{1}{|\mathcal{T}_+|} \sum_{t \in \mathcal{T}_+} J_t,$$

which emphasize behavior when faults actually occur.

All metrics are computed separately for the vertical and horizontal views. Final numbers are reported on the held-out test split described in §4.3.

# G CASE STUDIES

We visualize typical failure scenarios using Perfetto-rendered traces (Perfetto, 2025). Each case highlights distinct performance pathologies observed in our dataset.

## G.1 GPU MEMORY OVERFLOW

Memory-related operations such as `cuMemcpyAsync` exhibit execution times that increase from a few milliseconds under normal conditions to tens of milliseconds, often accompanied by interruptions in GPU kernel execution, as shown in Fig. 2. This behavior arises when the batch size of inference tasks exceeds the GPU memory capacity. Memory overflow triggers frequent swapping or failures in copy operations, and an early indication is a noticeable increase in the duration of `cuMemcpyAsync` events.

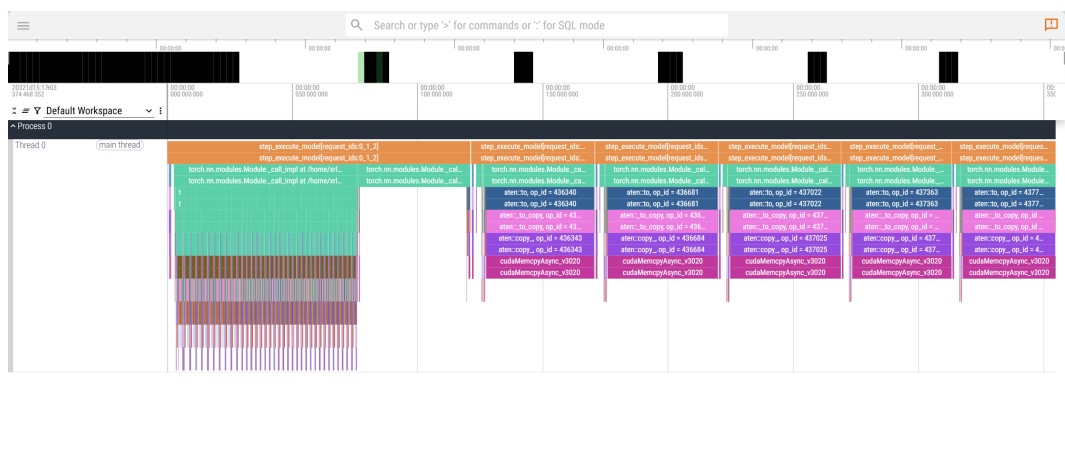

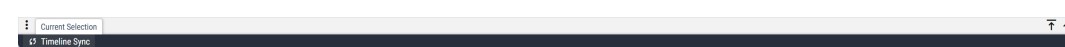

Figure 2: Perfetto trace for GPU memory overflow.

## G.2 COMPUTATION LATENCY

Front-end operators such as `torch.nn.funtional.linear` show nested back-end operators such as `aten::mm` with inflated execution times, leading to significant overall latency, as shown in Fig. 3. In abnormal cases, operator execution times in the PyTorch backend increase substantially, extending from the normal 1–5 ms range to more than 10 ms.

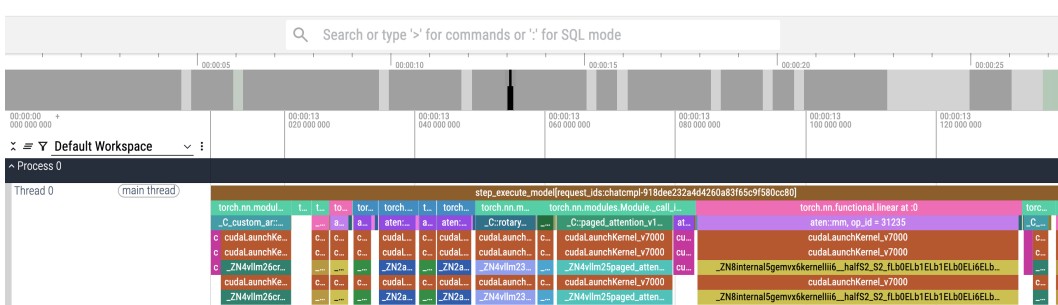

Figure 3: Perfetto trace for computation latency.

### G.3 COMMUNICATION LATENCY

Communication events such as `ncclAllReduce` become prominent in horizontal traces and are accompanied by increased waiting time at the network request layer. When abnormal latency occurs in distributed communication, execution times of some events grow markedly. For example, the duration of `ncclBroadcast` extends from microseconds under normal conditions to milliseconds, as shown in Fig. 4.

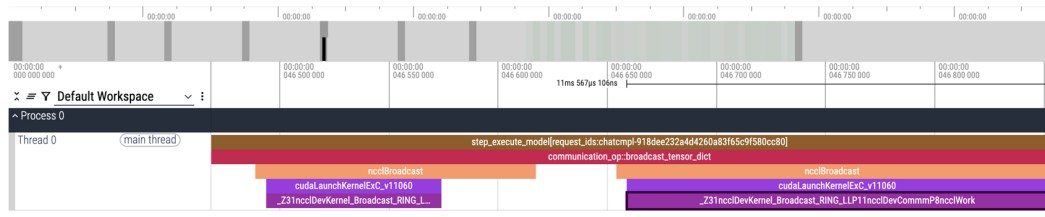

Figure 4: Perfetto trace for communication latency.

### G.4 INFERENCE ENGINE SCHEDULING DELAY

The execution time of `step_execute_model` is prolonged by several multiples compared with the normal baseline, as shown in Fig. 5. This anomaly often originates from errors in the internal scheduler of `vLLM` that trigger re-scheduling and introduce delays. Irregular nesting during the batch merging phase can interrupt request ID associations and impair concurrency and scheduling efficiency across multiple requests. This indicates potential issues in scheduling algorithms or state management within the inference engine.

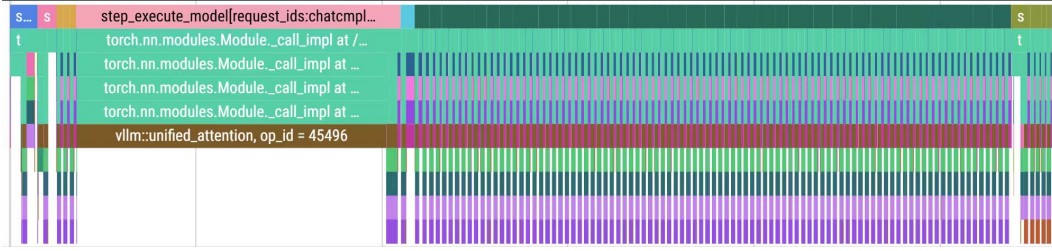

Figure 5: Perfetto trace for inference engine scheduling delay.

Finally, we note that TRUFFLD is not perfect. In rare cases it may miss or mis-localize anomalies. For example, when a particular operator family appears only a handful of times, Stage I GMM has limited statistics and may under-estimate its anomaly score, so subtle slowdowns can be treated as normal. Conversely, when two neighboring operators have highly correlated timings and similar semantics (e.g., a fused kernel and its immediate consumer), Stage II may attribute the anomaly to the wrong one within that small group. In heavily contended scenarios, upstream queueing or scheduling artifacts can also mask the true root cause and lead to ambiguous attributions. We include the above representative error cases to make the limitations of TRUFFLD explicit and to guide future extensions, such as better handling of rare operators and stronger temporal priors.

# H OVERHEAD

We quantify the runtime and resource overhead of the tracing pipeline by comparing executions with tracing disabled and enabled. Table 3 reports means across repeated runs. End-to-end wall time increases by about 9.8% and CPU utilization rises by 8.8%. System CPU time grows by 68.2% because the tracer records kernel and runtime events. Memory footprint and minor page faults increase slightly, and swap activity is unchanged. Overall, the non-intrusive tracing design provides rich observability at a cost that remains acceptable for production diagnosis and performance analysis.

Table 3: Runtime and resource overhead of tracing compared w/o tracing.

| Metric | No tracing (mean) | With tracing (mean) | Relative change |
|---|---|---|---|
| User CPU time (s) | 121.78 | 136.05 | +11.7% |
| System CPU time (s) | 19.13 | 32.20 | +68.2% |
| Wall time (s) | 64.77 | 71.15 | +9.8% |
| CPU utilization (%) | 217 | 236 | +8.8% |
| Max resident memory (KB) | 6,898,208 | 7,122,208 | +3.25% |
| Minor page faults (count) | 7,616,716 | 7,887,402 | +3.55% |
| Swaps (count) | 0 | 0 | no change |

# I ABLATION STUDIES

We ran ablation studies of our model. Table 4 reports an ablation on the two-stage anomaly detection pipeline. Using only Stage I GMM or only Stage II LLM reasoning clearly underperforms the full pipeline on both step-level and operator-level metrics in both views, confirming that numeric filtering and structural semantic reasoning are complementary rather than interchangeable.

Table 4: Ablation on the two-stage anomaly detection pipeline of TRUFFLD. *Full* uses both Stage I GMM and Stage II LLM reasoning. *Stage I only* removes the LLM and thresholds on GMM scores. *Stage II only* removes the GMM and lets the LLM decide directly from structured summaries.

| Method | Step | | | Operator Macro | | | Operator Macro$_+$ | | |
|---|---|---|---|---|---|---|---|---|---|
| | Acc. | Prec. | F1 | Prec. | F1 | Jaccard | Prec.$_+$ | F1$_+$ | Jaccard$_+$ |
| *Horizontal view* | | | | | | | | | |
| TRUFFLD (*Full*) | **0.926** | **0.961** | **0.922** | **0.435** | **0.880** | **0.842** | **0.898** | **0.789** | **0.711** |
| TRUFFLD (*Stage I only*) | 0.494 | 0.470 | 0.603 | 0.163 | 0.223 | 0.163 | 0.355 | 0.488 | 0.355 |
| TRUFFLD (*Stage II only*) | 0.617 | 0.632 | 0.668 | 0.221 | 0.303 | 0.215 | 0.403 | 0.521 | 0.380 |
| *Vertical view* | | | | | | | | | |
| TRUFFLD (*Full*) | **0.893** | **1.000** | **0.881** | **0.392** | **0.727** | **0.662** | **0.793** | **0.448** | **0.315** |
| TRUFFLD (*Stage I only*) | 0.549 | 0.545 | 0.644 | 0.149 | 0.219 | 0.149 | 0.288 | 0.422 | 0.288 |
| TRUFFLD (*Stage II only*) | 0.711 | 0.756 | 0.702 | 0.196 | 0.305 | 0.210 | 0.313 | 0.430 | 0.291 |

## J  SENSITIVITY TO GMM HYPERPARAMETERS

We examine the sensitivity of TRUFFLD to Stage I GMM hyperparameters on both horizontal and vertical views. We vary the number of mixture components $K$, the aggregation quantile $q$, and the rule for selecting central components, and report step-level F1 and operator-level macro F1. As shown in Tables 5 and 6, across a broad, reasonable range of settings, all metrics change only slightly around the default configuration. Only an intentionally extreme configuration leads to a noticeable drop, indicating that our default choice lies in a robust region.

Table 5: Sensitivity of TRUFFLD to GMM hyperparameters on the **horizontal** view.

| Setting | $K$ | $q$ | Central components rule | Step F1 (H) | Op Macro F1 (H) |
|---|---|---|---|---|---|
| Default | 4 | 0.25 | cum. mixture mass $\geq 0.8$ | 0.922 | 0.880 |
| $K$=2 (fewer components) | 2 | 0.25 | cum. mass $\geq 0.8$ | 0.921 | 0.874 |
| $K$=8 (more components) | 8 | 0.25 | cum. mass $\geq 0.8$ | 0.923 | 0.882 |
| Lower quantile ($q = 0.10$) | 4 | 0.10 | cum. mass $\geq 0.8$ | 0.918 | 0.876 |
| Higher quantile ($q = 0.50$) | 4 | 0.50 | cum. mass $\geq 0.8$ | 0.920 | 0.878 |
| Stricter centrality (top-1 component only) | 4 | 0.25 | most central component only | 0.915 | 0.868 |
| Extreme setting ($K = 1$, $q = 0.05$) | 1 | 0.05 | single Gaussian | 0.802 | 0.745 |

Table 6: Sensitivity of TRUFFLD to GMM hyperparameters on the **vertical** view.

| Setting | $K$ | $q$ | Central components rule | Step F1 (V) | Op Macro F1 (V) |
|---|---|---|---|---|---|
| Default | 4 | 0.25 | cum. mixture mass $\geq 0.8$ | 0.881 | 0.727 |
| $K$=2 (fewer components) | 2 | 0.25 | cum. mass $\geq 0.8$ | 0.879 | 0.720 |
| $K$=8 (more components) | 8 | 0.25 | cum. mass $\geq 0.8$ | 0.882 | 0.729 |
| Lower quantile ($q = 0.10$) | 4 | 0.10 | cum. mass $\geq 0.8$ | 0.878 | 0.722 |
| Higher quantile ($q = 0.50$) | 4 | 0.50 | cum. mass $\geq 0.8$ | 0.880 | 0.725 |
| Stricter centrality (top-1 component only) | 4 | 0.25 | most central component only | 0.874 | 0.716 |
| Extreme setting ($K = 1$, $q = 0.05$) | 1 | 0.05 | single Gaussian | 0.752 | 0.610 |

## K  DISCUSSION ON GENERALIZATION

We additionally ran TRUFFLD on SGLang, a Torch FSDP-based serving stack, and two vLLM versions (v0, v1). As shown in Table 7, we observe consistently high step-level and operator-level performance on both horizontal (cross-node) and vertical (within-node) views, which supports the claim that TRUFFLD is engine- and model-agnostic once CUDA + NVTX/CUPTI are available.

Table 7: Generalization of TRUFFLD across different inference engines and model stacks on both horizontal (cross-node) and vertical (within-node) views.

| | Step | | | Operator Macro | | | Operator Macro$_+$ | | |
|---|---|---|---|---|---|---|---|---|---|
| Method | Acc. | Prec. | F1 | Prec. | F1 | Jaccard | Prec.$_+$ | F1$_+$ | Jaccard$_+$ |
| *Horizontal view* | | | | | | | | | |
| TRUFFLD (SGLang) | 0.912 | 0.929 | 0.908 | 0.355 | 0.944 | 0.940 | 0.859 | 0.869 | 0.859 |
| TRUFFLD (Torch FSDP) | 0.910 | 0.925 | 0.907 | 0.356 | 0.945 | 0.942 | 0.864 | 0.873 | 0.864 |
| TRUFFLD (vLLM v0) | 0.905 | 0.919 | 0.901 | 0.358 | 0.947 | 0.944 | 0.870 | 0.879 | 0.870 |
| TRUFFLD (vLLM v1) | 0.915 | 0.927 | 0.912 | 0.356 | 0.945 | 0.941 | 0.861 | 0.871 | 0.861 |
| *Vertical view* | | | | | | | | | |
| TRUFFLD (SGLang) | 0.908 | 0.924 | 0.904 | 0.357 | 0.946 | 0.942 | 0.864 | 0.873 | 0.864 |
| TRUFFLD (Torch FSDP) | 0.914 | 0.924 | 0.912 | 0.359 | 0.949 | 0.945 | 0.870 | 0.881 | 0.870 |
| TRUFFLD (vLLM v0) | 0.910 | 0.926 | 0.907 | 0.354 | 0.944 | 0.939 | 0.857 | 0.867 | 0.857 |
| TRUFFLD (vLLM v1) | 0.913 | 0.928 | 0.909 | 0.359 | 0.949 | 0.944 | 0.870 | 0.880 | 0.870 |

## L   DISCUSSION ON STAGE I NUMERIC BACKBONE

In Stage I we adopt per-family/per-view Gaussian Mixture Models (GMMs) as the numeric backbone for three main reasons. First, GMMs are data-efficient and simple. They are easy to fit on relatively small per-operator datasets and naturally capture multi-modal normal behavior, which is common in LLM serving (e.g., different batch sizes, prefill vs. decode). Second, following classic density-based anomaly detection work (Xu et al., 2025; Qureshi et al., 2023), we model normal executions as forming dense clusters in a high-dimensional feature space, with anomalies as low-density outliers. Mixture models are a natural way to approximate this geometry, and GMM posteriors provide well-calibrated numeric confidences that can be directly used as anomaly scores. Third, in practice GMMs are cheap to train and update, and behave stably across operators and deployments, which is important for long-running systems.

To verify that GMM is a good design choice rather than an arbitrary one, we replaced Stage I with several alternative density models while keeping everything else (call-chain representation, Stage II LLM, fusion rule) identical: (i) a single Gaussian per family (one component), (ii) Kernel Density Estimation (KDE), (iii) Isolation Forest as a tree-based outlier detector, and (iv) a small normalizing flow model. Table 8 reports step-level F1 and operator-level macro F1 under each choice.

Table 8: Stage I density model vs. end-to-end performance of TRUFFLD. We report step-level F1 and operator-level macro F1 on both horizontal (H) and vertical (V) views.

| Stage I density model | Step F1 (H) | Step F1 (V) | Op Macro F1 (H) | Op Macro F1 (V) |
|---|---|---|---|---|
| **GMM (ours)** | **0.922** | **0.881** | **0.880** | **0.727** |
| Single Gaussian | 0.802 | 0.752 | 0.745 | 0.610 |
| KDE | 0.835 | 0.790 | 0.768 | 0.635 |
| Isolation Forest | 0.820 | 0.775 | 0.755 | 0.622 |
| Normalizing flow | 0.890 | 0.850 | 0.842 | 0.709 |

GMM provides the best or near-best F1 across both views, while being substantially lighter than KDE and normalizing flows (which incur higher training and inference cost). Single Gaussian and Isolation Forest are simpler but consistently underperform GMM. These results support our choice of GMM as a good balance between accuracy, complexity, and operational robustness for Stage I.

## M  DISCUSSION ON STAGE II LLM

**Cost of LLM.** For each Stage II reasoning call, we use on average 2,048 tokens, which corresponds to about \$0.000028 under a representative pricing. Even for a monitoring session with 5,000 steps, the total Stage II LLM cost is only around \$0.14, which is negligible compared to the GPU cost of serving the underlying LLM.

Table 9: Approximate cost of Stage II LLM reasoning. We assume an average of 2,048 tokens per call.

| Scenario | # Steps | Tokens / step | Total tokens | LLM cost (USD) |
|---|---|---|---|---|
| Single step (one horizontal or vertical view) | 1 | 2,048 | 2,048 | $\approx 2.8 \times 10^{-5}$ |
| Monitoring session | 5,000 | 2,048 | 10,240,000 | $\approx 0.14$ |

**Choice of LLM.** We also swapped the Stage II LLM with six alternatives (gpt-5-mini, doubao-seed-1-6-flash, claude-haiku-4-5, gemini-2.0-flash-lite, glm-4-flash) while keeping Stage I GMM unchanged. As shown in Table 10, all variants obtain very similar metrics, indicating that TRUFFLD is not sensitive to the specific LLM choice, thanks to the numeric filtering and candidate selection performed in Stage I.

Table 10: Effect of different LLMs in the Stage II reasoning on TRUFFLD. All variants share the same Stage I GMM, only the LLM used for structured reasoning is changed.

| Method | Step | | | Operator Macro | | | Operator Macro$_+$ | | |
|---|---|---|---|---|---|---|---|---|---|
| | Acc. | Prec. | F1 | Prec. | F1 | Jaccard | Prec.$_+$ | F1$_+$ | Jaccard$_+$ |
| *Horizontal view* | | | | | | | | | |
| TRUFFLD (gpt-5-nano) | 0.926 | 0.961 | 0.922 | 0.435 | 0.880 | 0.842 | 0.898 | 0.789 | 0.711 |
| TRUFFLD (gpt-5-mini) | 0.929 | 0.963 | 0.926 | 0.440 | 0.886 | 0.848 | 0.902 | 0.784 | 0.705 |
| TRUFFLD (claude-haiku-4-5) | 0.923 | 0.954 | 0.919 | 0.438 | 0.882 | 0.843 | 0.901 | 0.794 | 0.716 |
| TRUFFLD (gemini-2.0-flash-lite) | 0.918 | 0.953 | 0.914 | 0.432 | 0.874 | 0.836 | 0.893 | 0.781 | 0.704 |
| TRUFFLD (doubao-seed-1-6-flash) | 0.924 | 0.959 | 0.920 | 0.441 | 0.883 | 0.846 | 0.895 | 0.792 | 0.713 |
| TRUFFLD (glm-4-flash) | 0.921 | 0.958 | 0.917 | 0.437 | 0.878 | 0.840 | 0.900 | 0.787 | 0.709 |
| *Vertical view* | | | | | | | | | |
| TRUFFLD (gpt-5-nano) | 0.893 | 1.000 | 0.881 | 0.392 | 0.727 | 0.662 | 0.793 | 0.448 | 0.315 |
| TRUFFLD (gpt-5-mini) | 0.897 | 0.998 | 0.884 | 0.398 | 0.732 | 0.667 | 0.790 | 0.458 | 0.323 |
| TRUFFLD (claude-haiku-4-5) | 0.890 | 0.995 | 0.878 | 0.395 | 0.729 | 0.664 | 0.796 | 0.455 | 0.320 |
| TRUFFLD (gemini-2.0-flash-lite) | 0.886 | 0.992 | 0.874 | 0.388 | 0.721 | 0.656 | 0.787 | 0.442 | 0.309 |
| TRUFFLD (doubao-seed-1-6-flash) | 0.890 | 0.997 | 0.879 | 0.394 | 0.730 | 0.665 | 0.789 | 0.446 | 0.313 |
| TRUFFLD (glm-4-flash) | 0.889 | 0.996 | 0.878 | 0.390 | 0.725 | 0.660 | 0.792 | 0.450 | 0.317 |

