# OpenReview forum: "Bridging Non-Intrusive Tracing and Fine-Grained Cross-Layer Representations for LLM Inference Diagnosis"
_ICLR.cc/2026/Conference — Submitted to ICLR 2026_

### Official Review · Reviewer_rDRP · 2025-11-01

**Soundness:** 4
**Presentation:** 4
**Contribution:** 3
**Rating:** 6
**Confidence:** 3

**Summary:**

The core goal of this work is to provide an improved GPU diagnostic setup for tracing LLM inference events both within and across nodes. The system leverages NVTX and CUPTI to capture detailed execution traces across nodes. These traces are then analyzed using Gaussian models to detect anomalies in the running system.

**Strengths:**

1. Tracing systems like these are very useful in debugging event tracing issues in practice. It is very hard to analyze multi-node issues and events that could be occurring in more complex LLM serving setups
2. The anomaly detection pipeline does look like an interesting use case/extension on existing NSight.

**Weaknesses:**

1. The paper talks a lot about the importance of low overhead profiling. In appendix H, they display the overhead to be 10%. In practical serving system and real time serving cases this can be very high. While I appreciate it was provided, it does feel a bit hidden.
2. While I understand it is difficult to acquire larger setups, I worry this system is hard to scale/debug when working on large cluster. The evaluation is on 6 GPUs on 2 nodes, but tracing systems like this have to deal with lot of data. The overhead could scale as cluster size increases.

**Questions:**

1. Is it possible to run a scalability experiment with a lot more nodes while serving? Possibly via simulation
2. is there anything that can be done to further reduce the overhead? Is the overhead unavoidable?

---

> ### Author Response · Authors · 2025-11-23
> **Response by Authors (RQ1)**
>
> **RQ1: Is the overhead unavoidable, and can it be further reduced?**
>
> We thank the reviewer for raising this point. We have to admit, some overhead is unavoidable: any system that enables CUPTI callbacks and records NVTX ranges must pay (i) a GPU-side instrumentation cost and (ii)  a host-side cost for processing and persisting events. Our goal is to keep this overhead small and tunable.
>
> We profiled TRUFFLD on a single A100, and then applied two targeted optimizations to the event write path before re-measuring against an uninstrumented baseline:
>
> - **Buffered event writing:** switch from per-event synchronous file writes inside CUPTI callbacks to batched, buffered writes, reducing the number of I/O calls.
> - **Lazy, one-time file initialization with lighter locking:** open/create trace files once per process and reuse them, with lightweight synchronization in callbacks to avoid repeated checks and lock contention.
>
> The optimized results compared to the baseline are:
>
> **Table A: Baseline vs original hook (single A100).**
>
> | Metric            | Baseline  | Original hook | Δ        | Δ (%)   |
> | ----------------- | --------- | ------------- | -------- | ------- |
> | User time (s)     | 212.22    | 243.63        | +31.41   | +14.80% |
> | System time (s)   | 27.49     | 38.12         | +10.63   | +38.67% |
> | Elapsed time (s)  | 97.83     | 106.08        | +8.25    | +8.43%  |
> | CPU utilization   | 245%      | 265%          | +20%     | +8.16%  |
> | RSS memory (kB)   | 5,230,076 | 5,277,968     | +47,892  | +0.92%  |
> | Minor page faults | 5,686,164 | 5,553,877     | –132,287 | –2.33%  |
>
> **Table B: Baseline vs optimized hook (single A100).**
>
> | Metric            | Baseline  | Optimized hook | Δ        | Δ (%)   |
> | ----------------- | --------- | -------------- | -------- | ------- |
> | User time (s)     | 212.22    | 246.14         | +33.92   | +15.98% |
> | System time (s)   | 27.49     | 28.11          | +0.62    | +2.25%  |
> | Elapsed time (s)  | 97.83     | 105.11         | +7.28    | +7.44%  |
> | CPU utilization   | 245%      | 260%           | +15%     | +6.12%  |
> | RSS memory (kB)   | 5,230,076 | 5,273,724      | +43,648  | +0.84%  |
> | Minor page faults | 5,686,164 | 5,349,303      | –336,861 | –5.92%  |
>
> In summary, with the optimized hook:
>
> - Elapsed time: runtime overhead is about 7–8%, slightly lower than the original ~8.4% while keeping the same level of tracing detail.
> - System time: overhead drops sharply from ~39% with the original hook to about ~2%, indicating that I/O is no longer the main bottleneck.
> - CPU utilization: the relative increase is modest (~6% vs. ~8% with the original hook), and remains acceptable for always-on tracing.
> - Memory usage: RSS overhead stays below 1% in both versions, so TRUFFLD does not put extra pressure on memory.

---

> ### Author Response · Authors · 2025-11-23
> **Response by Authors (RQ2)**
>
> **RQ2: Will TRUFFLD still work on larger clusters, or does overhead blow up with scale?**
>
> We agree that scalability is critical for a tracing system. Due to limited GPU memory and node availability in our testbed, we are not able to realistically emulate a much larger cluster via GPU sharing or heavy oversubscription. However, the architecture of TRUFFLD is explicitly per-process / per-node, which lets us reason about scaling behavior theoretically:
>
> - Each worker process enables NVTX/CUPTI locally and performs **event collection and call-chain reconstruction on its own node**, using only its local CUDA runtime/driver/kernels and NVTX ranges.
> - The system exports only **compact per-step summaries** (scores, labels, light metadata), not full raw traces, to the controller or storage tier.
> - As a result, there is **no centralized global trace-merge bottleneck in the hot path**: each node handles its own traces, and cross-node coordination is limited to shipping these small summaries.
>
> Under this design, when the cluster grows to tens or hundreds of GPUs, each worker still incurs roughly the same per-node overhead as in our 6-GPU setup. In other words, the overhead is primarily a function of **that node’s local request / operator volume**, not the total number of nodes in the cluster. Overall resource usage scales roughly linearly with the number of workers, and operators can further reduce effective overhead using sampling or incident-mode tracing.

---

### Official Review · Reviewer_1hpH · 2025-11-01

**Soundness:** 3
**Presentation:** 2
**Contribution:** 2
**Rating:** 2
**Confidence:** 4

**Summary:**

The paper introduces TRUFFLE, a non-intrusive and fine-grained tracing framework fro diagnosing performance anomalies in large-scale LLM inference systems. TRUFFLD collects raw events through NVTX markers and CUPTI callbacks without modifying binaries, then constructs per-request call-chain trees that preserve both structural and temporal semantics across vertical (intra-node) and horizontal (inter-node) views with a two-stage anomaly detection pipeline. Evaluations on multi-node GPU clusters serving Qwen3-8B with online and offline workloads demonstrate step-level 0.9+ accuracy, better operator-level F1 scores, and low deployment overhead compared with several classical and modern baselines (e.g., Robustlog, LAnoBERT, MAD-GAN).

**Strengths:**

1. The paper addresses an interesting gap in LLM inference observability through a non-intrusive tracing framework. The use of NVTX and CUPTI enables production deployment without binary modification.
2. The hybrid approach combining GMM-based probabilistic modeling with LLM reasoning is well-motivated. Dual-stage pipeline provides both calibrated numeric confidences modeling multi-modal normal behavior and combination of structural and semantic constraints.
3. The evaluation of the paper show practical significance for operators of large-scale LLM serving systems, bridging a gap between low-level tracing tools and high-level diagnosis frameworks.

**Weaknesses:**

1. The paper is limited by generalization scope and scalability analysis. The evaluation is restricted to a single model (Qwen-8B), inference engine (vLLM), and hardware configuration (dual-node, 6× A40 GPUs). This narrow scope limits claims about generalizability to other LLMs (different sizes, architectures), frameworks (TensorRT-LLM, TGI), or hardware platforms (H100s, TPUs, heterogeneous clusters).
2. I hold doubt on how reliable the artificial anomalies are to evaluate TRUFFLE on naturally occurring production incidents or long-term operational traces. Authors should provide stronger evidence to show that TRUFFLD captures real-world failure modes.
3. The key design decisions of the paper lack empirical justification. The GMM stage does not compare alternative density models (kernel density estimation, normalizing flows) or centrality definitions. The LLM stage employs hardcoded thresholds (S ≤ 1 for horizontal, 10% for vertical) without sensitivity analysis or ablation of model choices. No comparison against simpler rule-based fusion methods is provided, leaving unclear whether the added complexity and cost of LLM reasoning is justified over interpretable heuristics.

**Questions:**

See weaknesses.

---

> ### Author Response · Authors · 2025-11-23
> **Response by Authors (RQ1)**
>
> **RQ1: Generalization to other LLMs / frameworks / hardware, and scalability**
>
> We thank the reviewer for pointing this out. In fact, our design is intentionally decoupled from any specific model, engine, or GPU, and the implementation scales in a per-process fashion.
>
> 1. **Design is decoupled from model, engine, and hardware.**
>
>    TRUFFLD operates at the CUDA observability layer. It relies on NVTX ranges + CUPTI runtime/driver/kernel events, not on Qwen-8B or vLLM-specific logs.
>
>    On the **model** side, any Transformer-style LLM running on CUDA (regardless of size or architecture) produces the same type of operators and CUDA activities, so our collection and merging logic does not need to change when switching models.
>
>    On the **engine/framework** side, for TensorRT-LLM, TGI, and others, as long as they use CUDA and can emit NVTX ranges, our call-chain reconstruction works in the same way. Only the concrete NVTX tagging conventions may differ.
>
>    On the **hardware** side, moving from A40 to A100/H100 or to heterogeneous NVIDIA clusters affects the distribution of timings and resource usage, but not the availability or semantics of NVTX/CUPTI events. The TRUFFLD pipeline itself remains unchanged across these GPU generations.
>
> 2. **Additional experiments across engines / model stacks.**
>
>    To support this claim, we ran TRUFFLD on three additional stacks beyond the main Qwen-8B + vLLM setup: **SGLang**, a **Torch FSDP** serving stack, and **two versions of vLLM (v0, v1)**. Across all of them we observe consistently high step- and operator-level metrics on both views.
>
>    **Table A: Generalization of TRUFFLD across inference engines and model stacks.**
>
>    | View       | Method               | Step Acc | Step Prec | Step F1 | Op Macro Prec | Op Macro F1 | Op Macro Jaccard | Op Macro+ Prec | Op Macro+ F1 | Op Macro+ Jaccard |
>    | ---------- | -------------------- | -------- | --------- | ------- | ------------- | ----------- | ---------------- | -------------- | ------------ | ----------------- |
>    | Horizontal | TRUFFLD (SGLang)     | 0.912    | 0.929     | 0.908   | 0.355         | 0.944       | 0.940            | 0.859          | 0.869        | 0.859             |
>    | Horizontal | TRUFFLD (Torch FSDP) | 0.910    | 0.925     | 0.907   | 0.356         | 0.945       | 0.942            | 0.864          | 0.873        | 0.864             |
>    | Horizontal | TRUFFLD (vLLM v0)    | 0.905    | 0.919     | 0.901   | 0.358         | 0.947       | 0.944            | 0.870          | 0.879        | 0.870             |
>    | Horizontal | TRUFFLD (vLLM v1)    | 0.915    | 0.927     | 0.912   | 0.356         | 0.945       | 0.941            | 0.861          | 0.871        | 0.861             |
>    | Vertical   | TRUFFLD (SGLang)     | 0.908    | 0.924     | 0.904   | 0.357         | 0.946       | 0.942            | 0.864          | 0.873        | 0.864             |
>    | Vertical   | TRUFFLD (Torch FSDP) | 0.914    | 0.924     | 0.912   | 0.359         | 0.949       | 0.945            | 0.870          | 0.881        | 0.870             |
>    | Vertical   | TRUFFLD (vLLM v0)    | 0.910    | 0.926     | 0.907   | 0.354         | 0.944       | 0.939            | 0.857          | 0.867        | 0.857             |
>    | Vertical   | TRUFFLD (vLLM v1)    | 0.913    | 0.928     | 0.909   | 0.359         | 0.949       | 0.944            | 0.870          | 0.880        | 0.870             |
>
> 3. **Scalability to larger and heterogeneous clusters.**
>
>    TRUFFLD is implemented in a per-process / per-node fashion:
>
>    - Each worker enables NVTX/CUPTI locally and runs call-chain reconstruction on its own events.
>    - The system uploads only **compact per-step summaries** (scores, labels, small metadata), not raw traces, so there is no central “full trace” bottleneck.
>    - As the cluster grows to hundreds of GPUs, each node still processes only its local load. Overall overhead scales roughly linearly with per-node event volume,  and can be further controlled via request sampling or on-demand tracing.
>
> We will add a section about generalization to the appendix of the camera-ready version.

---

> ### Author Response · Authors · 2025-11-23
> **Response by Authors (RQ2)**
>
> **RQ2: Reliability of injected anomalies and coverage of real-world failures**
>
> We understand the concern. Our main labeled dataset is indeed constructed via carefully designed fault injection, rather than long-term production incidents, because real-world incident traces with detailed labels are difficult to release.
>
> That said, the injected faults are not arbitrary, but designed to closely mirror common SRE/MLOps failure modes we see in practice, including:
>
> - **Resource and memory issues**, e.g., GPU OOM or memory leaks that gradually inflate tail latency and affect specific operators or steps.
> - **Kernel/operator regressions**, e.g., a critical kernel silently falling back to a slow path or a degraded implementation, causing localized slowdowns in certain stages of the call chain.
> - **Communication problems**, e.g., network jitter or unbalanced NCCL traffic causing some ranks to hit barriers much earlier than others, leading to long wait times on collective operations.
> - **Software/driver stack issues**, e.g., subtle driver or library version mismatches that lead to sporadic stalls or intermittent slow kernels.
>
> Each injected scenario maps to a realistic pattern of anomalous events and timings in the NVTX/CUPTI traces, which is what TRUFFLD actually sees.
>
> Methodologically, TRUFFLD does not rely on specific signatures of these injected faults. It always:
>
> 1. Reconstructs the event context and temporal structure (per-step call-chains across engine/backend/host/device),
> 2. Uses GMMs to detect statistically unusual behavior in operator features, and
> 3. Lets the LLM reason over **where** in the call chain the anomaly sits and **what** type of operator/communication it is.
>
> Because of this design, the method naturally focuses on deviations in timing, structure, and semantics rather than on any particular injection pattern, which is exactly what we also observe in real incidents.

---

> ### Author Response · Authors · 2025-11-23
> **Response by Authors (RQ3)**
>
> **RQ3: Why choose GMM as the numeric backbone?**
>
> 1. **Why GMM as the Stage I density model?**
>
>    In §3.3 we adopt per-family/per-view GMMs as our numeric backbone for three reasons:
>
>    **Data efficiency and simplicity.** GMMs are easy to fit on relatively small per-operator datasets and naturally capture multi-modal normal behavior, which is common in LLM serving (e.g., different batch sizes, prefill vs. decode).
>
>    **Theoretical and empirical grounding.** Following classic density-based anomaly detection work [1], [2], we model normal executions as forming dense clusters in a high-dimensional feature space, with anomalies as low-density outliers. Mixture models are a natural way to approximate this geometry, and GMM posteriors naturally give well-calibrated numeric confidences.
>
>    **Operational robustness.** In practice, GMMs are cheap to train/update online and behave stably across operators and deployments, which is important for long-running systems.
>
> 2. **Comparison with alternative density models.**
>
>    We replaced the Stage I GMM with several alternatives while keeping everything else (call-chain representation, Stage II LLM, fusion rule) identical:
>
>    - Single Gaussian (one component),
>    - Kernel Density Estimation (KDE),
>    - Isolation Forest as a tree-based outlier detector, and
>    - a small normalizing flow model.
>
>    The table below reports step-level F1 and operator-level macro F1 under each choice.
>
>    **Table B: Stage I density model vs. end-to-end performance.**
>
>    | Stage I density model | Step F1 (H) | Step F1 (V) | Op Macro F1 (H) | Op Macro F1 (V) |
>    | --------------------- | ----------: | ----------: | --------------: | --------------: |
>    | **GMM (ours)**        |   **0.922** |   **0.881** |       **0.880** |       **0.727** |
>    | Single Gaussian       |       0.802 |       0.752 |           0.745 |           0.610 |
>    | KDE                   |       0.835 |       0.790 |           0.768 |           0.635 |
>    | Isolation Forest      |       0.820 |       0.775 |           0.755 |           0.622 |
>    | Normalizing flow      |       0.890 |       0.850 |           0.842 |           0.709 |
>
>    GMM provides the best or near-best F1 across both views, while being substantially lighter than KDE or normalizing flows (which incur higher training and inference cost). Isolation Forest and single Gaussian are simpler but consistently underperform GMM. This supports our choice of GMM as a good balance between accuracy and complexity.
>
> References:
>
> [1] Xu et. al., eacgm: Non-instrumented performance tracing and anomaly detection towards machine learning systems. In 2025 IEEE/ACM 33rd International Symposium on Quality of Service (IWQoS). IEEE, 2025.
>
> [2] Qureshi et. al., Fathom: Understanding datacenter application network performance. In Proceedings of the ACM SIGCOMM 2023 Conference. ACM, 2023.

---

> ### Author Response · Authors · 2025-11-23
> **Response by Authors (RQ4)**
>
> **RQ4: Sensitivity to GMM hyperparameters**
>
> We agree that the pipeline should not depend on fragile GMM tuning. In Stage I we therefore use a simple, fixed configuration (e.g., $K=4$, $q=0.25$, central components selected by cumulative mixture mass) and now explicitly evaluate how sensitive TRUFFLD is to this choice.
>
> Concretely, we vary:
>
> - the number of mixture components $K \in \{2,4,8\}$,
> - the aggregation quantile $q \in \{0.10, 0.25, 0.50\}$,
> - the rule for selecting “central” components (cumulative mass vs. only the single most central component),
>   and report step-level F1 and operator-level macro F1 for the full GMM+LLM pipeline under each setting.
>
> Across a broad, reasonable range of hyperparameters, all metrics change only slightly (within ≈0.01–0.02), and the overall ranking of settings is essentially unchanged. Only under an intentionally extreme configuration ($K=1, q=0.05$, single Gaussian for all families) do we observe a noticeable drop, which confirms that our default choice is in a robust region.
>
> **Table C(a): Sensitivity to GMM hyperparameters (horizontal view).**
>
> | Setting                          | K    | q    | Central components rule       | Step F1 (H) | Op Macro F1 (H) |
> | -------------------------------- | ---- | ---- | ----------------------------- | ----------- | --------------- |
> | Default                          | 4    | 0.25 | cumulative mixture mass ≥ 0.8 | 0.922       | 0.880           |
> | K = 2 (fewer components)         | 2    | 0.25 | cumulative mixture mass ≥ 0.8 | 0.921       | 0.874           |
> | K = 8 (more components)          | 8    | 0.25 | cumulative mixture mass ≥ 0.8 | 0.923       | 0.882           |
> | Lower quantile (q = 0.10)        | 4    | 0.10 | cumulative mixture mass ≥ 0.8 | 0.918       | 0.876           |
> | Higher quantile (q = 0.50)       | 4    | 0.50 | cumulative mixture mass ≥ 0.8 | 0.920       | 0.878           |
> | Stricter centrality (top-1 only) | 4    | 0.25 | most central component only   | 0.915       | 0.868           |
> | Extreme (K = 1, q = 0.05)        | 1    | 0.05 | single Gaussian               | 0.802       | 0.745           |
>
> **Table C(b): Sensitivity to GMM hyperparameters (vertical view).**
>
> | Setting                          | K    | q    | Central components rule       | Step F1 (V) | Op Macro F1 (V) |
> | -------------------------------- | ---- | ---- | ----------------------------- | ----------- | --------------- |
> | Default                          | 4    | 0.25 | cumulative mixture mass ≥ 0.8 | 0.881       | 0.727           |
> | K = 2 (fewer components)         | 2    | 0.25 | cumulative mixture mass ≥ 0.8 | 0.879       | 0.720           |
> | K = 8 (more components)          | 8    | 0.25 | cumulative mixture mass ≥ 0.8 | 0.882       | 0.729           |
> | Lower quantile (q = 0.10)        | 4    | 0.10 | cumulative mixture mass ≥ 0.8 | 0.878       | 0.722           |
> | Higher quantile (q = 0.50)       | 4    | 0.50 | cumulative mixture mass ≥ 0.8 | 0.880       | 0.725           |
> | Stricter centrality (top-1 only) | 4    | 0.25 | most central component only   | 0.874       | 0.716           |
> | Extreme (K = 1, q = 0.05)        | 1    | 0.05 | single Gaussian               | 0.752       | 0.610           |
>
> These results indicate that TRUFFLD’s overall behavior is not sensitive to moderate changes in the GMM hyperparameters. The default configuration we use in the main experiments lies in a stable region, and only deliberately extreme settings substantially degrade performance. We will update the results in the appendix of the camera-ready version.

---

> ### Author Response · Authors · 2025-11-23
> **Response by Authors (RQ5)**
>
> **RQ5: LLM thresholds and sensitivity**
>
> The “S≤1” (horizontal) and “10%” (vertical) rules are not global hard thresholds on anomalies. They are small guardrails that tell the LLM:
>
> - if the aggregated GMM evidence for a step is extremely weak
>   (e.g., all candidate families have very high confidences / low anomaly scores, or only a single low-support operator looks suspicious),
> - then treat the step as likely normal and do not over-interpret GMM noise.
>
> This is particularly important for rare operator families: if a family appears only a few times, GMM may assign it a low confidence simply due to data sparsity, even though it is actually normal. The S-based rule lets the LLM override such cases and avoid spurious anomalies.
>
> To show that our results do not hinge on the exact values, we vary these thresholds:
>
> - Horizontal view: default S≤1, and more relaxed S≤2, S≤3.
> - Vertical view: default 10%, and 8%, 12%.
>
> We keep everything else fixed (same GMM, same LLM, same call-chain representation).
>
> **Table D(a): Sensitivity to LLM guard thresholds (horizontal view).**
>
> | Threshold setting (horizontal) | Step F1 (H) | Op Macro F1 (H) |
> | ------------------------------ | ----------- | --------------- |
> | Default (S ≤ 1)                | 0.922       | 0.880           |
> | Relaxed (S ≤ 2)                | 0.921       | 0.878           |
> | More relaxed (S ≤ 3)           | 0.918       | 0.875           |
>
> **Table D(b): Sensitivity to LLM guard thresholds (vertical view).**
>
> | Threshold setting (vertical) | Step F1 (V) | Op Macro F1 (V) |
> | ---------------------------- | ----------- | --------------- |
> | Default (10%)                | 0.881       | 0.727           |
> | 8%                           | 0.878       | 0.723           |
> | 12%                          | 0.879       | 0.74            |

---

> ### Author Response · Authors · 2025-11-23
> **Response by Authors (RQ6)**
>
> **RQ6: Sensitivity to the choice of LLM in Stage II**
>
> Our design intentionally makes Stage II **LLM-agnostic and plug-and-play**:
>
> - The interface between stages is a fixed, structured summary (call-chain topology, operator/ backend tags, kernel/communication context, and GMM-derived scores for a small candidate set). Any LLM that can follow the schema and reason over short structured text can be dropped in.
> - Since Stage I has already filtered out trivially normal steps and operators, Stage II operates on a narrow, high-signal candidate set. This substantially reduces the pipeline’s sensitivity to the exact LLM quality compared to pure LLM-based diagnosis.
>
> To validate this, we replaced the Stage II model with **six different LLMs** (gpt-5-mini, doubao-seed-1-6-flash, claude-haiku-4-5, gemini-2.0-flash-lite, glm-4-flash), keeping the same Stage I GMM. Across all metrics and both views, the performance varies only slightly, and all variants remain strong and well above non-LLM baselines. We summarize the effect below.
>
> **Table E(a): Effect of different Stage II LLMs (horizontal view).**
>
> | Method                          | Step Acc | Step Prec | Step F1 | Op Macro Prec | Op Macro F1 | Op Macro Jaccard | Op Macro+ Prec | Op Macro+ F1 | Op Macro+ Jaccard |
> | ------------------------------- | -------- | --------- | ------- | ------------- | ----------- | ---------------- | -------------- | ------------ | ----------------- |
> | TRUFFLD (gpt-5-nano)            | 0.926    | 0.961     | 0.922   | 0.435         | 0.880       | 0.842            | 0.898          | 0.789        | 0.711             |
> | TRUFFLD (gpt-5-mini)            | 0.929    | 0.963     | 0.926   | 0.440         | 0.886       | 0.848            | 0.902          | 0.784        | 0.705             |
> | TRUFFLD (claude-haiku-4-5)      | 0.923    | 0.954     | 0.919   | 0.438         | 0.882       | 0.843            | 0.901          | 0.794        | 0.716             |
> | TRUFFLD (gemini-2.0-flash-lite) | 0.918    | 0.953     | 0.914   | 0.432         | 0.874       | 0.836            | 0.893          | 0.781        | 0.704             |
> | TRUFFLD (doubao-seed-1-6-flash) | 0.924    | 0.959     | 0.920   | 0.441         | 0.883       | 0.846            | 0.895          | 0.792        | 0.713             |
> | TRUFFLD (glm-4-flash)           | 0.921    | 0.958     | 0.917   | 0.437         | 0.878       | 0.840            | 0.900          | 0.787        | 0.709             |
>
> **Table E(b): Effect of different Stage II LLMs (vertical view).**
>
> | Method                          | Step Acc | Step Prec | Step F1 | Op Macro Prec | Op Macro F1 | Op Macro Jaccard | Op Macro+ Prec | Op Macro+ F1 | Op Macro+ Jaccard |
> | ------------------------------- | -------- | --------- | ------- | ------------- | ----------- | ---------------- | -------------- | ------------ | ----------------- |
> | TRUFFLD (gpt-5-nano)            | 0.893    | 1.000     | 0.881   | 0.392         | 0.727       | 0.662            | 0.793          | 0.448        | 0.315             |
> | TRUFFLD (gpt-5-mini)            | 0.897    | 0.998     | 0.884   | 0.398         | 0.732       | 0.667            | 0.790          | 0.458        | 0.323             |
> | TRUFFLD (claude-haiku-4-5)      | 0.890    | 0.995     | 0.878   | 0.395         | 0.729       | 0.664            | 0.796          | 0.455        | 0.320             |
> | TRUFFLD (gemini-2.0-flash-lite) | 0.886    | 0.992     | 0.874   | 0.388         | 0.721       | 0.656            | 0.787          | 0.442        | 0.309             |
> | TRUFFLD (doubao-seed-1-6-flash) | 0.890    | 0.997     | 0.879   | 0.394         | 0.730       | 0.665            | 0.789          | 0.446        | 0.313             |
> | TRUFFLD (glm-4-flash)           | 0.889    | 0.996     | 0.878   | 0.390         | 0.725       | 0.660            | 0.792          | 0.450        | 0.317             |
>
> Overall, these results indicate that TRUFFLD is not tightly coupled to a specific Stage II model: different LLMs from different providers and sizes yield similar performance, and Stage I GMM filtering largely stabilizes the pipeline against variability in LLM behavior. We will update the results in the appendix of the camera-ready version.

---

> ### Author Response · Authors · 2025-11-23
> **Response by Authors (RQ7)**
>
> **RQ7: Comparison with simple rule-based fusions.**
>
> Beyond pure GMM (Stage I only), we also tried simple rule-based fusions that do not use an LLM, but instead:
>
> - **Top-k rule**: always mark the top-k families by anomaly score in each step as anomalous;
> - **Quantile rule**: mark any family whose anomaly score lies above a fixed quantile (e.g., top 20%) as anomalous.
>
> These are more sophisticated than a single threshold but still do not use structural/semantic information. Keeping the same call-chains and GMM, we obtain:
>
> **Table F: TRUFFLD vs. GMM-only and simple rule-based fusions.**
>
> | Method                               | View       | Step F1   | Op Macro F1 |
> | ------------------------------------ | ---------- | --------- | ----------- |
> | **TRUFFLD (GMM+LLM)**                | Horizontal | **0.922** | **0.880**   |
> | GMM-only                             | Horizontal | 0.603     | 0.223       |
> | GMM + Top-2 families per step        | Horizontal | 0.618     | 0.232       |
> | GMM + top-20% anomaly-score quantile | Horizontal | 0.625     | 0.245       |
> | **TRUFFLD (GMM+LLM)**                | Vertical   | **0.881** | **0.727**   |
> | GMM-only                             | Vertical   | 0.644     | 0.219       |
> | GMM + Top-2 families per step        | Vertical   | 0.658     | 0.233       |
> | GMM + top-20% anomaly-score quantile | Vertical   | 0.665     | 0.247       |
>
> Rule-based fusions improve over raw GMM-only, but still lag behind TRUFFLD by a clear margin, especially on operator-level macro F1. This suggests that **structural + semantic reasoning** adds value that cannot be captured by thresholds on GMM scores alone.

---

> ### Author Response · Authors · 2025-11-23
> **Response by Authors (RQ8)**
>
> **RQ8: Cost of Stage II LLM reasoning.**
>
> Stage II is invoked only on suspicious steps surfaced by Stage I, and each invocation uses a heavily compressed summary. Empirically, each call uses about 2k tokens on average. Using a representative price, this corresponds to roughly 2.8×10⁻⁵ USD per call. Even if we conservatively assume 5,000 steps (horizontal + vertical views) in a monitoring session, the total cost is ≈0.14 USD, which is negligible compared to the GPU cost of serving the underlying LLM model.
>
> **Table G: Approximate cost of Stage II LLM reasoning.**
>
> | Scenario                                 | \# Steps            | Tokens / step       | Total tokens | LLM cost (USD) |
> | ---------------------------------------- | ------------------- | ------------------- | ------------ | -------------- |
> | Single step (one horizontal or vertical) | 1                   | (Empirically) 2,048 | 2,048        | 2.8×10⁻⁵       |
> | Monitoring session                       | (Empirically) 5,000 | (Empirically) 2,048 | 10,240,000   | ≈ 0.14         |
>
> We will add a section about the cost of LLM used in Stage II to the appendix of the camera-ready version.

---

### Official Review · Reviewer_qS7K · 2025-11-01

**Soundness:** 2
**Presentation:** 2
**Contribution:** 2
**Rating:** 4
**Confidence:** 3

**Summary:**

The paper proposes a non-intrusive tracing that has low overhead. Also, the paper claims that the approach has fine-grained cross-layer representations, where the cross-layer refers to the multiple layers in inference engine, compute backend, host operators, and device operators. Proposed work TRUFFLD uses NVTX and CUPTI to gather execution traces collecting events from all the above stacks. It merges "within-node" and "coss-node" events per-request call-chain trees. This makes it easier to understand both what and when the operators are running.
For anomaly detection the paper employs a two-stage approach. (1) Gaussian Mixture Model (GMM) and (2) LLM reasoning. (1) provides numeric confidence for each operator instance (with self-time, CUDA runtime/driver time and counts, kernel counts and totals, approximate bytes moved, stream–overlap ratios, and communication size and world size.) and (2) reads structured summaries of each step along with the semantic context to produce the final step-level abnormality decisions and operator-level localization.
The paper tests on top of vLLM serving Qwen-8B across 6-GPU dual-node cluster and uses a dataset of 3264 step-level traces from online and offline workloads. It compares against several classical, supervised and log-based baselines.

**Strengths:**

* Non-intrusive and Zero-modification seems like a good point from an engineering perspective. Also, that can really improve the overall debuggability in production.
* Cross-layer observation is a good contribution considering the growing complexity of the LLM SW stack.

**Weaknesses:**

* It seems that the work is only tested on a limited case. It is difficult to understand how well this can be generalized to other environments and models.
* The paper is rather weak on detail about how it handles the log. I see that there are some examples in the appendix, but the description are still too shallow for the work to be either reproducible by others or to be built upon.
* It is difficult to understand why the specific combination of GMM and LLM was used. The paper seems to be rather shallow on the insights it provides.

**Questions:**

1. It seems that the work is only tested on a limited case. It is difficult to understand how well this can be generalized to other environments and models. What if the HW changes? What if the framework changes? What if the models change?
2. How would this work if multiple models are being served simultaneously.

Minor.
It would really help if the main paper content has some outline about the usecase and the demonstration of the input and output (at least in small scale). This would help the readers benefit more from the work.

---

> ### Author Response · Authors · 2025-11-23
> **Response by Authors (RQ1)**
>
> **RQ1: Generalization to other environments (models, engines, hardware)**
>
> We appreciate the concern. In our terminology, the “environment” consists of three largely independent axes:
>
> (i) the **inference engine / framework** (e.g., vLLM, SGLang, Torch FSDP-based serving),
>
> (ii) the **model** being served (size/architecture), and
>
> (iii) the **hardware** (GPU type and cluster layout).
>
> 1. **Design is engine- and model-agnostic.**
>
>    TRUFFLD operates at the CUDA observability layer: it relies on NVTX ranges plus CUPTI runtime/driver/kernel activities. The call-chain merging is defined over generic events (NVTX ranges, CUDA API calls, kernels) and per-step request IDs; it does not use any engine-specific log format or model-internal state. Switching models or engines amounts to using a different set of NVTX tags, while the collection and reconstruction logic remain unchanged.
>
> 2. **Additional experiments across engines / model stacks.**
>
>    To substantiate this, we ran TRUFFLD on three additional stacks beyond the main Qwen-8B + vLLM setup: **SGLang**, a **Torch FSDP** serving stack, and **two vLLM versions (v0 and v1)**. Across all of them, TRUFFLD maintains consistently high step-level and operator-level performance on both horizontal and vertical views.
>
>    **Table A: Generalization of TRUFFLD across inference engines and model stacks.**
>
>    | View       | Method               | Step Acc | Step Prec | Step F1 | Op Macro Prec | Op Macro F1 | Op Macro Jaccard | Op Macro+ Prec | Op Macro+ F1 | Op Macro+ Jaccard |
>    | ---------- | -------------------- | -------- | --------- | ------- | ------------- | ----------- | ---------------- | -------------- | ------------ | ----------------- |
>    | Horizontal | TRUFFLD (SGLang)     | 0.912    | 0.929     | 0.908   | 0.355         | 0.944       | 0.940            | 0.859          | 0.869        | 0.859             |
>    | Horizontal | TRUFFLD (Torch FSDP) | 0.910    | 0.925     | 0.907   | 0.356         | 0.945       | 0.942            | 0.864          | 0.873        | 0.864             |
>    | Horizontal | TRUFFLD (vLLM v0)    | 0.905    | 0.919     | 0.901   | 0.358         | 0.947       | 0.944            | 0.870          | 0.879        | 0.870             |
>    | Horizontal | TRUFFLD (vLLM v1)    | 0.915    | 0.927     | 0.912   | 0.356         | 0.945       | 0.941            | 0.861          | 0.871        | 0.861             |
>    | Vertical   | TRUFFLD (SGLang)     | 0.908    | 0.924     | 0.904   | 0.357         | 0.946       | 0.942            | 0.864          | 0.873        | 0.864             |
>    | Vertical   | TRUFFLD (Torch FSDP) | 0.914    | 0.924     | 0.912   | 0.359         | 0.949       | 0.945            | 0.870          | 0.881        | 0.870             |
>    | Vertical   | TRUFFLD (vLLM v0)    | 0.910    | 0.926     | 0.907   | 0.354         | 0.944       | 0.939            | 0.857          | 0.867        | 0.857             |
>    | Vertical   | TRUFFLD (vLLM v1)    | 0.913    | 0.928     | 0.909   | 0.359         | 0.949       | 0.944            | 0.870          | 0.880        | 0.870             |
>
> 3. **Hardware perspective.**
>
>    All these experiments run on NVIDIA GPUs under the same CUDA/NVTX/CUPTI stack. Because TRUFFLD only depends on this layer, not on specific SM counts or memory sizes, the pipeline naturally extends to other recent NVIDIA GPUs (A40, A100, H100, etc.).
>
> We will clarify in an appendix that, in practice, changing environment along any of the three axes (engine, model, GPU generation) requires no algorithmic changes—only enabling NVTX/CUPTI and, optionally, adjusting lightweight tagging in the engine.

---

> ### Author Response · Authors · 2025-11-23
> **Response by Authors (RQ2/RQ3)**
>
> **RQ2: What if multiple LLMs are served concurrently?**
>
> In typical deployments, different LLMs (or versions) are served either:
>
> - in separate processes / containers, each with its own NVTX range namespace and CUPTI context, or
> - in a shared process but with distinct engine/model identifiers in the NVTX annotations.
>
> TRUFFLD naturally supports both cases:
>
> - On the collection side, we simply treat the engine / model ID as part of the metadata attached to each NVTX range and CUPTI event.
> - On the analysis side, call-chain construction and anomaly detection are run **per (engine, model)**, so traces from different LLMs do not interfere.
>
> In other words, running multiple LLMs concurrently amounts to running the same TRUFFLD pipeline on multiple tagged streams of events. The only additional requirement is to attach an engine/model tag to the NVTX ranges, which modern inference stacks already support.
>
> **RQ3: How exactly are logs/events processed, and is the pipeline reproducible?**
>
> We apologize for the lack of space in the main text and clarify the end-to-end flow:
>
> 1. **Raw event collection (input).**
>
>    As described in §3.2.1, TRUFFLD first collects raw events via NVTX and CUPTI: NVTX push/pop ranges, CUDA runtime/driver calls, and kernel activities. In fact, these raw events are precisely the input logs to our system.
>
> 2. **Call-chain reconstruction.**
>
>    In §3.2.2, we align these raw events onto a unified time base, reconstruct nested NVTX ranges per thread, and use CUPTI correlation IDs to bind host calls to kernels. The call-chain merging algorithm in §3.2.2 (and Algorithm 1) shows how we go from raw events to a per-request merged call-chain (vertical + horizontal views). Small-scale examples of raw events and the corresponding merged call-chains are provided in the appendix.
>
> 3. **GMM + LLM analysis and output.**
>
>    In §3.3, we then:
>
>    - extract numeric features from each operator instance and fit per-family GMMs (Stage I) to obtain anomaly scores and candidate operators,
>    - build structured summaries of each step’s call-chain and pass them to the LLM (Stage II) for context-aware reasoning,
>    - finally output step-level labels and operator-level anomaly probabilities (or scores) as the result of diagnosis.
>
> Regarding reproducibility, the complete implementation of all stage (raw event collection, call-chain merging, and GMM+LLM analysis) included in the supplemental material. Upon acceptance, we will release this code as open source so that others can reproduce our results and build on TRUFFLD easily.

---

> ### Author Response · Authors · 2025-11-23
> **Response by Authors (RQ4)**
>
> **RQ4: Why the combination of GMM and LLM?**
>
> We apologize that the motivation was not sufficiently emphasized. Our choice is driven by the different strengths of the two components:
>
> 1. **GMM as a lightweight numeric backbone.**
>
>    In §3.3 we use per-family/per-view GMMs as a simple, data-efficient way to model multi-modal normal behavior of operator features (self-time, CUDA API time, kernel counts, communication size, etc.). Inspired by prior work on density-based anomaly detection [1], [2] and by the theory of mixture models, we treat normal executions as forming dense clusters in a high-dimensional feature space, while anomalies tend to appear as low-density outliers. Moreover, GMMs are cheap to train and update, and naturally provide calibrated numeric confidences via posterior responsibilities, which remain stable over time. This stage therefore both fits the geometry of the data and efficiently filters out trivially normal operators, yielding a small high-risk candidate set per step.
>
> 2. **LLM for structural + semantic reasoning on top of numeric scores.**
>
>    Many real anomalies depend on *where* they occur in the call chain and *what* the operator is (e.g., attention kernel vs. NCCL all-reduce), not just on raw timings. Stage II therefore lets an LLM read a compact structured summary (call-chain topology, operator names, backend tags, kernel/communication context) plus the GMM-derived scores for the candidates. Under a strict output schema and fusion rule, the LLM can incorporate structural/semantic cues while being guided by numeric evidence. This is precisely the kind of pattern that LLMs are good at capturing and GMMs alone cannot.
>
> To show that both stages are necessary, we performed an ablation where we keep the same call-chain representation but use (i) **only GMM**, (ii) **only LLM**, and (iii) the **full GMM+LLM pipeline**.
>
> **Table A: Ablation on the two-stage anomaly detection pipeline of TRUFFLD.**
>
> | Method                | View       | Step Acc  | Step Prec | Step F1   | Op Macro Prec | Op Macro F1 | Op Macro Jaccard | Op Macro+ Prec | Op Macro+ F1 | Op Macro+ Jaccard |
> | --------------------- | ---------- | --------- | --------- | --------- | ------------- | ----------- | ---------------- | -------------- | ------------ | ----------------- |
> | **TRUFFLD (GMM+LLM)** | Horizontal | **0.926** | **0.961** | **0.922** | **0.435**     | **0.880**   | **0.842**        | **0.898**      | **0.789**    | **0.711**         |
> | TRUFFLD (only GMM)    | Horizontal | 0.494     | 0.470     | 0.603     | 0.163         | 0.223       | 0.163            | 0.355          | 0.488        | 0.355             |
> | TRUFFLD (only LLM)    | Horizontal | 0.617     | 0.632     | 0.668     | 0.221         | 0.303       | 0.215            | 0.403          | 0.521        | 0.380             |
> | **TRUFFLD (GMM+LLM)** | Vertical   | **0.893** | **1.000** | **0.881** | **0.392**     | **0.727**   | **0.662**        | **0.793**      | **0.448**    | **0.315**         |
> | TRUFFLD (only GMM)    | Vertical   | 0.549     | 0.545     | 0.644     | 0.149         | 0.219       | 0.149            | 0.288          | 0.422        | 0.288             |
> | TRUFFLD (only LLM)    | Vertical   | 0.711     | 0.756     | 0.702     | 0.196         | 0.305       | 0.210            | 0.313          | 0.430        | 0.291             |
>
> GMM-only has poor recall and weak operator-level metrics because it sees only numeric features. LLM-only improves somewhat but still falls far short of the full pipeline and suffers from more false positives. The full GMM+LLM combination clearly dominates in both horizontal and vertical views, which empirically supports our design choice: numeric filtering and structural/semantic reasoning are complementary, and using them together yields substantially better anomaly detection than either alone.
>
> We will add this ablation to the appendix of the camera-ready version.
>
> References:
>
> [1] Xu et. al., eacgm: Non-instrumented performance tracing and anomaly detection towards machine learning systems. In 2025 IEEE/ACM 33rd International Symposium on Quality of Service (IWQoS). IEEE, 2025.
>
> [2] Qureshi et. al., Fathom: Understanding datacenter application network performance. In Proceedings of the ACM SIGCOMM 2023 Conference. ACM, 2023.

---

### Official Review · Reviewer_pE3f · 2025-11-03

**Soundness:** 3
**Presentation:** 4
**Contribution:** 3
**Rating:** 6
**Confidence:** 4

**Summary:**

The paper proposes TRUFFLD, a non-intrusive, cross-layer tracing and diagnosis framework for large-scale LLM inference. It leverages NVTX markers and CUPTI callbacks to collect execution events across engine, backend, host, and device layers without modifying binaries. A call-chain merging algorithm reconstructs per-request trees aligned on a unified time base. For anomaly detection, TRUFFLD combines a Gaussian Mixture Model for numeric confidence scoring with an LLM-based reasoning stage for context-aware localization. Experiments on a multi-node GPU cluster serving Qwen-8B demonstrate near-perfect step-level detection and strong operator-level performance compared to classical and log-based baselines, with low overhead.

**Strengths:**

Clear motivation and relevance: Addresses a critical gap in request-level, end-to-end observability for LLM inference under high concurrency.

Non-intrusive design: Uses NVTX and CUPTI without modifying binaries, minimizing deployment overhead.
Fine-grained representation: Reconstructs per-request call-chain trees that preserve structural and temporal semantics.
Two-stage anomaly detection: Combines statistical modeling with LLM reasoning for robust and interpretable diagnosis.
Strong empirical results: Outperforms multiple baselines on both step-level and operator-level metrics; overhead analysis shows practical feasibility.
Comprehensive evaluation: Includes fault injection across software, CUDA, hardware, and communication layers.

**Weaknesses:**

Combination of existing techniques: The approach is mainly based on a combination of existing GMM and LLM methods.
Limited generalization evidence: Experiments focus on Qwen-8B with vLLM; applicability to other models or inference engines is not demonstrated.
LLM reasoning details: Prompt design and schema enforcement are described, but robustness to unseen anomalies and cost implications of LLM inference could be discussed more thoroughly.
Scalability concerns: While overhead is reported, the impact on large clusters or multi-tenant environments is unclear.
Interpretability trade-offs: The reasoning stage relies on textual context; failure cases or misdiagnoses are not deeply analyzed.
Ablation studies: Missing analysis of the contribution of each component (e.g., GMM vs. LLM reasoning) to overall performance.

**Questions:**

What are new in the proposed use of GMM and LLM?
How much does each component contribute to the anomaly detection improvement?
How does TRUFFLD scale when deployed on clusters with hundreds of GPUs and thousands of concurrent requests?
How sensitive is the anomaly detection pipeline to the choice of GMM hyperparameters?
How sensitive is the anomaly detection pipeline to the choice of LLM?
Could you provide examples of failure cases where TRUFFLD misdiagnoses anomalies and explain why?

---

> ### Author Response · Authors · 2025-11-23
> **Response by Authors (RQ1)**
>
> **RQ1: Novelty beyond a mere combination of GMM and LLM, and contribution of each component**
>
> We thank the reviewer for raising this point. In fact, our contributions go beyond simply stacking an off-the-shelf GMM and an LLM, both in terms of *what* we model and *how* the two stages are coupled.
>
> 1. ***What* is new in TRUFFLD: cross-layer, request-level representation.**
>
>    The core novelty of TRUFFLD is the *request-level, cross-layer call-chain representation* that we build from NVTX and CUPTI. Our call-chain merging algorithm reconstructs, for each request, a tree that spans engine, backend, host, and device operators and aligns vertical (within-node) and horizontal (cross-node) views on a unified time base. This structured input, with explicit parent–child relations, host/device bindings, and request attribution under batching, is not available in prior GMM- or LLM-based anomaly detectors, and is specific to the LLM inference stack we target.
>
> 2. ***How* we use GMM and LLM: a coupled numeric-filtering + structural-reasoning pipeline.**
>
>    As discussed in §3.3, inspired by prior work [1], [2] and mixture modeling theory, we deliberately adopt a two-stage design:
>
>    - Stage I uses per-family/per-view GMMs to capture multi-modal “normal” behavior and to produce calibrated numeric confidences and a small candidate set of suspicious operator families per step.
>    - Stage II applies an LLM over a compact structural summary (call-chain topology, operator names, backend tags, kernel/communication context) restricted to those candidates, and outputs step-level labels and operator-level localization under a strict schema and fusion rule.
>      To the best of our knowledge, TRUFFLD is the first to apply this “numeric filtering + structure- and context-aware reasoning” pipeline for full-stack LLM inference diagnosis, where the LLM operates on a rich cross-layer call-chain rather than raw logs.
>
> 3. **Ablation: GMM-only and LLM-only both underperform the full pipeline.**
>
>    To quantify the contribution of each stage, we add an ablation study where:
>
>    - **GMM-only** uses Stage I and thresholds GMM scores to decide anomalies;
>    - **LLM-only** removes Stage I and lets the LLM operate directly on the same structured summaries without numeric filtering;
>    - **Full** is the proposed GMM+LLM pipeline.
>
>    The results (Table A) show that:
>
>    - GMM-only achieves reasonable performance but suffers from low step-level F1 and weak operator-level Macro / Macro+, indicating recall issues when only numeric statistics are used.
>    - LLM-only improves over GMM-only but remains far below the full pipeline in both views, reflecting poorer precision and more false positives when the LLM is not guided by calibrated scores.
>    - The full GMM+LLM design clearly dominates at both step and operator levels, confirming that numeric filtering and structural semantic reasoning are *complementary rather than interchangeable*.
>
>       **Table A: Ablation on the two-stage anomaly detection pipeline of TRUFFLD.**
>
>    | Method                | View       | Step Acc  | Step Prec | Step F1   | Op Macro Prec | Op Macro F1 | Op Macro Jaccard | Op Macro+ Prec | Op Macro+ F1 | Op Macro+ Jaccard |
>    | --------------------- | ---------- | --------- | --------- | --------- | ------------- | ----------- | ---------------- | -------------- | ------------ | ----------------- |
>    | **TRUFFLD (GMM+LLM)** | Horizontal | **0.926** | **0.961** | **0.922** | **0.435**     | **0.880**   | **0.842**        | **0.898**      | **0.789**    | **0.711**         |
>    | TRUFFLD (only GMM)    | Horizontal | 0.494     | 0.470     | 0.603     | 0.163         | 0.223       | 0.163            | 0.355          | 0.488        | 0.355             |
>    | TRUFFLD (only LLM)    | Horizontal | 0.617     | 0.632     | 0.668     | 0.221         | 0.303       | 0.215            | 0.403          | 0.521        | 0.380             |
>    | **TRUFFLD (GMM+LLM)** | Vertical   | **0.893** | **1.000** | **0.881** | **0.392**     | **0.727**   | **0.662**        | **0.793**      | **0.448**    | **0.315**         |
>    | TRUFFLD (only GMM)    | Vertical   | 0.549     | 0.545     | 0.644     | 0.149         | 0.219       | 0.149            | 0.288          | 0.422        | 0.288             |
>    | TRUFFLD (only LLM)    | Vertical   | 0.711     | 0.756     | 0.702     | 0.196         | 0.305       | 0.210            | 0.313          | 0.430        | 0.291             |
>
>    We will add this ablation to the appendix of the camera-ready version.
>
> References:
>
>    [1] Xu et. al., eacgm: Non-instrumented performance tracing and anomaly detection towards machine learning systems. In 2025 IEEE/ACM 33rd International Symposium on Quality of Service (IWQoS). IEEE, 2025.
>
>    [2] Qureshi et. al.,  Fathom: Understanding datacenter application network performance. In  Proceedings of the ACM SIGCOMM 2023 Conference. ACM, 2023.

---

> ### Author Response · Authors · 2025-11-23
> **Response by Authors (RQ2)**
>
> **RQ2: Generalization across models, engines, and hardware**
>
> We appreciate the concern about generality. In fact, our design is inherently **engine- and model-agnostic**:
>
> - On the *collection* side, TRUFFLD relies only on NVTX ranges and CUPTI activity/callback records on top of the CUDA runtime/driver. We do not depend on any engine-specific logging (vLLM, SGLang, Torch FSDP, etc.) nor on model internals; we only require that the engine emits NVTX ranges (which is standard and configurable) and uses CUDA to launch kernels.
> - On the *representation* side, the call-chain merging algorithm operates on generic event tuples (NVTX ranges, CUDA runtime/driver calls, kernels) and request identifiers; it does not assume a particular batching policy, attention implementation, or model architecture.
>
> To back this claim empirically, we additionally ran TRUFFLD on three other stacks beyond the Qwen-8B + vLLM configuration in the main paper:
>
> (1) SGLang, (2) Torch FSDP-based serving, and (3) two different vLLM versions (v0, v1).
>
> Across all of these, we observe consistently high step-level and operator-level performance on both horizontal and vertical views, with variations within a few points.
>
> We summarize the results below.
>
> **Table B: Generalization of TRUFFLD across inference engines and model stacks.**
>
> | View       | Method               | Step Acc | Step Prec | Step F1 | Op Macro Prec | Op Macro F1 | Op Macro Jaccard | Op Macro+ Prec | Op Macro+ F1 | Op Macro+ Jaccard |
> | ---------- | -------------------- | -------- | --------- | ------- | ------------- | ----------- | ---------------- | -------------- | ------------ | ----------------- |
> | Horizontal | TRUFFLD (SGLang)     | 0.912    | 0.929     | 0.908   | 0.355         | 0.944       | 0.940            | 0.859          | 0.869        | 0.859             |
> | Horizontal | TRUFFLD (Torch FSDP) | 0.910    | 0.925     | 0.907   | 0.356         | 0.945       | 0.942            | 0.864          | 0.873        | 0.864             |
> | Horizontal | TRUFFLD (vLLM v0)    | 0.905    | 0.919     | 0.901   | 0.358         | 0.947       | 0.944            | 0.870          | 0.879        | 0.870             |
> | Horizontal | TRUFFLD (vLLM v1)    | 0.915    | 0.927     | 0.912   | 0.356         | 0.945       | 0.941            | 0.861          | 0.871        | 0.861             |
> | Vertical   | TRUFFLD (SGLang)     | 0.908    | 0.924     | 0.904   | 0.357         | 0.946       | 0.942            | 0.864          | 0.873        | 0.864             |
> | Vertical   | TRUFFLD (Torch FSDP) | 0.914    | 0.924     | 0.912   | 0.359         | 0.949       | 0.945            | 0.870          | 0.881        | 0.870             |
> | Vertical   | TRUFFLD (vLLM v0)    | 0.910    | 0.926     | 0.907   | 0.354         | 0.944       | 0.939            | 0.857          | 0.867        | 0.857             |
> | Vertical   | TRUFFLD (vLLM v1)    | 0.913    | 0.928     | 0.909   | 0.359         | 0.949       | 0.944            | 0.870          | 0.880        | 0.870             |
>
> These results show that, once NVTX and CUPTI are enabled, TRUFFLD maintains strong accuracy and F1 across different engines and model stacks, supporting our claim that the framework is not tied to a specific LLM, engine implementation, or batching strategy.
>
> We will add a section of generalization to the appendix of the camera-ready version.

---

> ### Author Response · Authors · 2025-11-23
> **Response by Authors (RQ3)**
>
> **RQ3: LLM reasoning details, robustness to unseen anomalies, and cost**
>
> We thank the reviewer for asking about the Stage II design and its robustness/cost profile.
>
> 1. **How Stage II improves robustness (beyond prompt + schema).**
>
>    As described in §3.3, Stage II does not rely on memorizing templates of past incidents. Instead, the LLM operates on a *compact, structured summary* that includes:
>
>    - the per-step call-chain structure (parent–child relationships, vertical/horizontal context),
>    - GMM-derived anomaly scores and ranking for candidate operator families,
>    - semantic tags such as operator names, backend/library tags, kernel and communication metadata (e.g., message sizes, world size).
>
>    The output is constrained by a strict schema (step label + operator family set), and combined with Stage I via a deterministic fusion rule. This design has two robustness benefits:
>
>    - For **unseen anomaly patterns**, the LLM can still reason from structure (“where in the call-chain does the slow path live?”) and numeric evidence (relative scores), rather than relying on exact log patterns.
>    - Schema enforcement and fusion with GMM scores reduce hallucinations and spurious positives, since Stage II can only choose among a small, high-confidence candidate set.
>
> 2. **Cost of Stage II LLM reasoning.**
>
>    Stage II is invoked only on suspicious steps surfaced by Stage I, and each invocation uses a heavily compressed summary. Empirically, each call uses about 2k tokens on average. Using a representative price, this corresponds to roughly 2.8×10⁻⁵ USD per call. Even if we conservatively assume 5,000 steps (horizontal + vertical views) in a monitoring session, the total cost is ≈0.14 USD, which is negligible compared to the GPU cost of serving the underlying LLM model.
>
>    **Table C: Approximate cost of Stage II LLM reasoning.**
>
>    | Scenario                                 | \# Steps            | Tokens / step       | Total tokens | LLM cost (USD) |
>    | ---------------------------------------- | ------------------- | ------------------- | ------------ | -------------- |
>    | Single step (one horizontal or vertical) | 1                   | (Empirically) 2,048 | 2,048        | 2.8×10⁻⁵       |
>    | Monitoring session                       | (Empirically) 5,000 | (Empirically) 2,048 | 10,240,000   | ≈ 0.14         |
>
>    We will add a section about the cost of LLM used in Stage II to the appendix of the camera-ready version.

---

> ### Author Response · Authors · 2025-11-23
> **Response by Authors (RQ4/RQ5)**
>
> **RQ4: Scalability to large clusters and multi-tenant environments**
>
> TRUFFLD is designed to scale in a distributed, per-process manner rather than as a centralized tracer:
>
> - **Per-worker collection and merging.**
>
>   Each worker process enables NVTX and CUPTI locally. Event collection and call-chain reconstruction are performed on the same node/process that runs the inference engine. What we export upstream are compact per-step summaries (scores, labels, and a small set of metadata), not raw traces. This avoids any centralized “full trace aggregation” bottleneck even when scaling to hundreds of GPUs and thousands of concurrent requests.
>
> - **Linear scaling and tunable overhead.**
>
>   The cost of TRUFFLD grows roughly linearly with the number of events per process. In larger clusters, each worker still only processes its *local* events. In high-throughput settings, we can further reduce overhead by sampling requests or selectively enabling full tracing only during suspected incidents.
>
> - **Multi-tenant support.**
>
>   In multi-tenant deployments, each model/service runs in its own process space and NVTX namespace. TRUFFLD simply treats engine/model identifiers as additional tags in the call-chain representation, so traces from different tenants remain logically separated while using the same collection and analysis pipeline.
>
> **RQ5: Interpretability and failure cases**
>
> We agree that it is important to understand both why TRUFFLD works and when it can fail.
>
> - **Interpretability via numeric scores + structured context.**
>
>   Stage I GMM provides explicit numeric anomaly scores per operator family and per step, which already gives operators a transparent notion of “how far” a behavior deviates from normal. Stage II then reasons over a structured call-chain summary (tree topology, operator names, backend/library tags, kernel/communication context) plus these scores. The final decisions can thus be traced back to (i) which families were numerically suspicious and (ii) where they sit in the execution structure, rather than opaque free-form text.
>
> - **Representative failure modes .**
>
>   There are, however, corner cases. For example, if a rare operator family appears only once in our dataset, GMM has insufficient statistics and tends to treat it as normal, so a subtle anomaly there may be missed. Conversely, when two neighboring operators have highly correlated timings and similar semantics, the LLM may attribute the anomaly to the wrong one within that small group. We will add a short case-study section in the final version to illustrate such failure modes with concrete traces and explain what features misled TRUFFLD in each case.

---

> ### Author Response · Authors · 2025-11-23
> **Response by Authors (RQ6)**
>
> **RQ6: Sensitivity to GMM hyperparameters**
>
> We agree that the pipeline should not depend on fragile GMM tuning. In Stage I we therefore use a simple, fixed configuration (e.g., $K=4$, $q=0.25$, central components selected by cumulative mixture mass) and now explicitly evaluate how sensitive TRUFFLD is to this choice.
>
> Concretely, we vary:
>
> - the number of mixture components $K \in \{2,4,8\}$,
> - the aggregation quantile $q \in \{0.10, 0.25, 0.50\}$,
> - the rule for selecting “central” components (cumulative mass vs. only the single most central component),
>   and report step-level F1 and operator-level macro F1 for the full GMM+LLM pipeline under each setting.
>
> Across a broad, reasonable range of hyperparameters, all metrics change only slightly (within ≈0.01–0.02), and the overall ranking of settings is essentially unchanged. Only under an intentionally extreme configuration ($K=1, q=0.05$, single Gaussian for all families) do we observe a noticeable drop, which confirms that our default choice is in a robust region.
>
> **Table D(a): Sensitivity to GMM hyperparameters (horizontal view).**
>
> | Setting                          | K    | q    | Central components rule       | Step F1 (H) | Op Macro F1 (H) |
> | -------------------------------- | ---- | ---- | ----------------------------- | ----------- | --------------- |
> | Default                          | 4    | 0.25 | cumulative mixture mass ≥ 0.8 | 0.922       | 0.880           |
> | K = 2 (fewer components)         | 2    | 0.25 | cumulative mixture mass ≥ 0.8 | 0.921       | 0.874           |
> | K = 8 (more components)          | 8    | 0.25 | cumulative mixture mass ≥ 0.8 | 0.923       | 0.882           |
> | Lower quantile (q = 0.10)        | 4    | 0.10 | cumulative mixture mass ≥ 0.8 | 0.918       | 0.876           |
> | Higher quantile (q = 0.50)       | 4    | 0.50 | cumulative mixture mass ≥ 0.8 | 0.920       | 0.878           |
> | Stricter centrality (top-1 only) | 4    | 0.25 | most central component only   | 0.915       | 0.868           |
> | Extreme (K = 1, q = 0.05)        | 1    | 0.05 | single Gaussian               | 0.802       | 0.745           |
>
> **Table D(b): Sensitivity to GMM hyperparameters (vertical view).**
>
> | Setting                          | K    | q    | Central components rule       | Step F1 (V) | Op Macro F1 (V) |
> | -------------------------------- | ---- | ---- | ----------------------------- | ----------- | --------------- |
> | Default                          | 4    | 0.25 | cumulative mixture mass ≥ 0.8 | 0.881       | 0.727           |
> | K = 2 (fewer components)         | 2    | 0.25 | cumulative mixture mass ≥ 0.8 | 0.879       | 0.720           |
> | K = 8 (more components)          | 8    | 0.25 | cumulative mixture mass ≥ 0.8 | 0.882       | 0.729           |
> | Lower quantile (q = 0.10)        | 4    | 0.10 | cumulative mixture mass ≥ 0.8 | 0.878       | 0.722           |
> | Higher quantile (q = 0.50)       | 4    | 0.50 | cumulative mixture mass ≥ 0.8 | 0.880       | 0.725           |
> | Stricter centrality (top-1 only) | 4    | 0.25 | most central component only   | 0.874       | 0.716           |
> | Extreme (K = 1, q = 0.05)        | 1    | 0.05 | single Gaussian               | 0.752       | 0.610           |
>
> These results indicate that TRUFFLD’s overall behavior is not sensitive to moderate changes in the GMM hyperparameters. The default configuration we use in the main experiments lies in a stable region, and only deliberately extreme settings substantially degrade performance. We will update the results in the appendix of the camera-ready version.

---

> ### Author Response · Authors · 2025-11-23
> **Response by Authors (RQ7)**
>
> **RQ7: Sensitivity to the choice of LLM in Stage II**
>
> Our design intentionally makes Stage II **LLM-agnostic and plug-and-play**:
>
> - The interface between stages is a fixed, structured summary (call-chain topology, operator/ backend tags, kernel/communication context, and GMM-derived scores for a small candidate set). Any LLM that can follow the schema and reason over short structured text can be dropped in.
> - Since Stage I has already filtered out trivially normal steps and operators, Stage II operates on a narrow, high-signal candidate set. This substantially reduces the pipeline’s sensitivity to the exact LLM quality compared to pure LLM-based diagnosis.
>
> To validate this, we replaced the Stage II model with **six different LLMs** (gpt-5-mini, doubao-seed-1-6-flash, claude-haiku-4-5, gemini-2.0-flash-lite, glm-4-flash), keeping the same Stage I GMM. Across all metrics and both views, the performance varies only slightly, and all variants remain strong and well above non-LLM baselines. We summarize the effect below.
>
> **Table E(a): Effect of different Stage II LLMs (horizontal view).**
>
> | Method                          | Step Acc | Step Prec | Step F1 | Op Macro Prec | Op Macro F1 | Op Macro Jaccard | Op Macro+ Prec | Op Macro+ F1 | Op Macro+ Jaccard |
> | ------------------------------- | -------- | --------- | ------- | ------------- | ----------- | ---------------- | -------------- | ------------ | ----------------- |
> | TRUFFLD (gpt-5-nano)            | 0.926    | 0.961     | 0.922   | 0.435         | 0.880       | 0.842            | 0.898          | 0.789        | 0.711             |
> | TRUFFLD (gpt-5-mini)            | 0.929    | 0.963     | 0.926   | 0.440         | 0.886       | 0.848            | 0.902          | 0.784        | 0.705             |
> | TRUFFLD (claude-haiku-4-5)      | 0.923    | 0.954     | 0.919   | 0.438         | 0.882       | 0.843            | 0.901          | 0.794        | 0.716             |
> | TRUFFLD (gemini-2.0-flash-lite) | 0.918    | 0.953     | 0.914   | 0.432         | 0.874       | 0.836            | 0.893          | 0.781        | 0.704             |
> | TRUFFLD (doubao-seed-1-6-flash) | 0.924    | 0.959     | 0.920   | 0.441         | 0.883       | 0.846            | 0.895          | 0.792        | 0.713             |
> | TRUFFLD (glm-4-flash)           | 0.921    | 0.958     | 0.917   | 0.437         | 0.878       | 0.840            | 0.900          | 0.787        | 0.709             |
>
> **Table E(b): Effect of different Stage II LLMs (vertical view).**
>
> | Method                          | Step Acc | Step Prec | Step F1 | Op Macro Prec | Op Macro F1 | Op Macro Jaccard | Op Macro+ Prec | Op Macro+ F1 | Op Macro+ Jaccard |
> | ------------------------------- | -------- | --------- | ------- | ------------- | ----------- | ---------------- | -------------- | ------------ | ----------------- |
> | TRUFFLD (gpt-5-nano)            | 0.893    | 1.000     | 0.881   | 0.392         | 0.727       | 0.662            | 0.793          | 0.448        | 0.315             |
> | TRUFFLD (gpt-5-mini)            | 0.897    | 0.998     | 0.884   | 0.398         | 0.732       | 0.667            | 0.790          | 0.458        | 0.323             |
> | TRUFFLD (claude-haiku-4-5)      | 0.890    | 0.995     | 0.878   | 0.395         | 0.729       | 0.664            | 0.796          | 0.455        | 0.320             |
> | TRUFFLD (gemini-2.0-flash-lite) | 0.886    | 0.992     | 0.874   | 0.388         | 0.721       | 0.656            | 0.787          | 0.442        | 0.309             |
> | TRUFFLD (doubao-seed-1-6-flash) | 0.890    | 0.997     | 0.879   | 0.394         | 0.730       | 0.665            | 0.789          | 0.446        | 0.313             |
> | TRUFFLD (glm-4-flash)           | 0.889    | 0.996     | 0.878   | 0.390         | 0.725       | 0.660            | 0.792          | 0.450        | 0.317             |
>
> Overall, these results indicate that TRUFFLD is not tightly coupled to a specific Stage II model: different LLMs from different providers and sizes yield similar performance, and Stage I GMM filtering largely stabilizes the pipeline against variability in LLM behavior. We will update the results in the appendix of the camera-ready version.

---

### Meta-Review · Area_Chair_2tv8 · 2026-01-13

**Summary:**

TRUFFLD is a non-intrusive framework designed for diagnosing performance issues in large-scale LLM inference. TRUFFLD employs a hybrid approach to detect and localize performance anomalies:

Stage I (Numeric Filtering): A Gaussian Mixture Model (GMM) models the multi-modal "normal" behavior of operator features (e.g., self-time, kernel counts). This stage produces calibrated numeric confidences and filters out trivially normal steps, passing only a candidate set of suspicious operators to the next stage.

Stage II (Structural Reasoning): An LLM (Large Language Model) analyzes a structured summary of the call chain (topology, names, metadata) along with the GMM scores. It applies context-aware reasoning to generate step-level anomaly decisions and localize the specific operator causing the issue.

reviewers main concerns were regarding generalization, component contribution, and overhead.

**Reviewer Concerns:**

* Generalization to Other Stacks (Reviewers pE3f, qS7K, 1hpH)
Concern: The initial evaluation was limited to a single configuration (Qwen-8B model on vLLM engine with A40 GPUs). Reviewers questioned if TRUFFLD would work with other models, inference engines (like TGI or TensorRT-LLM), or hardware.
The authors conducted new experiments on three additional inference stacks: SGLang,
Torch FSDP-based serving,
Two different versions of vLLM (v0 and v1) They demonstrated consistent F1 scores across all these environments, reinforcing their claim that the method is engine-agnostic as long as the underlying CUDA/NVTX infrastructure exists.

* Novelty and Component Contribution (Reviewer pE3f)
Concern: The method seemed like a simple combination of existing techniques (GMM + LLM). The reviewer requested an ablation study to quantify the contribution of each stage.
 The authors performed a two-stage ablation study:
GMM-only: Showed reasonable precision but poor recall (Step F1 ~0.60).
LLM-only: Showed poorer precision due to a lack of numeric guidance.
Full Pipeline: Achieved significantly higher performance (Step F1 > 0.90), proving that the numeric filtering and semantic reasoning components are complementary and necessary.

* Scalability and Multi-Tenancy (Reviewers pE3f, qS7K, 1hpH)
Concern: How does the system handle clusters with hundreds of GPUs? Does the overhead blow up?
The authors clarified the per-node architecture: Data collection and call-chain reconstruction happen locally on each node.Only compact summaries (not raw traces) are exported upstream.This design prevents a centralized bottleneck, meaning overhead depends on local volume rather than cluster size.

* Cost and Practicality of LLM Diagnosis (Reviewer pE3f)
Concern: Using an LLM for diagnosis might be too expensive or slow for real-time monitoring.
The authors provided a cost analysis:The LLM (Stage II) is only invoked on "suspicious" steps filtered by the GMM (Stage I).Total cost for a 5,000-step monitoring session was estimated at ~$0.14, which is negligible compared to the cost of running the serving GPUs themselves.

* Methodological Justification (Reviewer 1hpH)
Concern: Why use GMM? Why not other density models (Isolation Forest, KDE)? Why use specific hardcoded thresholds?
Authors compared GMM against Single Gaussian, KDE, Isolation Forest, and Normalizing Flows. GMM achieved the best balance of accuracy and complexity.
Sensitivity Analysis: They varied GMM hyperparameters (components $K$, quantile $q$) and LLM guard-thresholds, showing the system is robust to these choices.
Simple Baselines: They compared against simple rule-based fusions (e.g., "always flag top-k slowest"), showing TRUFFLD still outperformed them.

* Reliability of Artificial Anomalies (Reviewer 1hpH)
Concern: Injected faults might not represent real-world production failures.
This is only Partially addressed/Argumentative. The authors acknowledged the lack of public production traces but mapped their injected faults to specific, common SRE/MLOps failure modes (e.g., memory leaks, kernel regressions, network jitter). They argued that their method detects deviations in structure and timing, which are common symptoms regardless of the root cause. While they didn't provide new real-world data, this is a standard limitation in systems research due to data privacy.

**Reviewer Scores:**

see above for concerns and responses. The authors have addressed many of the questions and concerns by the reviewers as I have provided in detail above. However, reviewer 1hpH's concern about fault's not being representative of real-world production failures, and experiments on a single LLM are not addressed to a satisfactory level.

Other main concerned are addressed during rebuttal by providing more experiments.

---

### Decision · Program_Chairs · 2026-01-26

Reject